# Robustifying Learning-Augmented Caching Efficiently without Compromising 1-Consistency

**Peng Chen**[1]    **Hailiang Zhao**[1*]    **Jiaji Zhang**[1]    **Xueyan Tang**[2]
**Yixuan Wang**[3]    **Shuiguang Deng**[1*]

[1]Zhejiang University    [2]Nanyang Technological University
[3]Nanjing University of Aeronautics and Astronautics

{pgchen,hliangzhao,zhangjiaji,dengsg}@zju.edu.cn
asxytang@ntu.edu.sg    wangyixuan@nuaa.edu.cn

## Abstract

The online caching problem aims to minimize cache misses when serving a sequence of requests under a limited cache size. While naive learning-augmented caching algorithms achieve ideal 1-consistency, they lack robustness guarantees. Existing robustification methods either sacrifice 1-consistency or introduce excessive computational overhead. In this paper, we introduce GUARD, a lightweight robustification framework that enhances the robustness of a broad class of learning-augmented caching algorithms to $2H_{k-1} + 2$, while preserving their 1-consistency. GUARD achieves the current best-known trade-off between consistency and robustness, with only $\mathcal{O}(1)$ additional per-request overhead, thereby maintaining the original time complexity of the base algorithm. Extensive experiments across multiple real-world datasets and prediction models validate the effectiveness of GUARD in practice.

## 1 Introduction

The classical *caching* (or *paging*) problem is a fundamental online optimization problem with widespread applications in operating systems, web caching, and database management. It involves serving a sequence of $n$ page requests using a cache of limited size $k$ ($1 \leq k \ll \infty$). A request incurs zero cost if the corresponding page is already in the cache (*cache hit*); otherwise, a *cache miss* occurs, requiring the page to be loaded into the cache, possibly evicting another page to make space. The objective is to minimize the total number of cache misses over the entire request sequence.

Caching has been extensively studied in both deterministic and randomized settings. In the offline setting where future requests are known, Belady [1] proposed the first optimal strategy, *Belady's rule*, which evicts the page whose next request lies furthest in the future. In the online setting, where future requests are unknown, strong theoretical lower bounds exist. Specifically, no deterministic algorithm can achieve a competitive ratio better than $k$ [2], and no randomized algorithm can do better than $H_k$ [3], where $H_k = \sum_{i=1}^{k} \frac{1}{i}$ is the $k$-th harmonic number satisfying $\ln(k+1) \leq H_k \leq \ln(k) + 1$. Several algorithms have been developed to approach these bounds: MARKER [3] achieves a competitive ratio of $2H_k - 1$, while EQUITABLE [4] matches the optimal $H_k$, but at the cost of significantly higher time complexity $\mathcal{O}(k^2)$ per request, limiting its practicality.

Recent advances in machine learning have inspired a new paradigm: *learning-augmented algorithms*, which leverage predictive models to guide decision-making in online problems such as caching. These algorithms aim to improve performance when predictions are accurate while remaining robust

---

[*]Corresponding authors: hliangzhao@zju.edu.cn, dengsg@zju.edu.cn

in the face of prediction errors. Their performance is typically evaluated using three key metrics: *consistency*, *robustness*, and *smoothness*. Consistency measures the competitive ratio under perfect predictions; Robustness bounds the algorithm performance under arbitrary prediction errors, and smoothness captures how the competitive ratio degrades as prediction errors increase.

This paper focuses on learning-augmented algorithm design for the classical caching problem, which lies at the intersection of machine learning and theoretical computer science. While the caching problem itself is fundamental and pervasive in computer systems, our work demonstrates the potential for safely integrating machine learning into cache systems.

## 1.1 Related Work

The first learning-augmented caching algorithm is BLINDORACLE [5], which blindly follows predictions by evicting the page predicted to be requested furthest in the future. While this yields ideal 1-consistency (a competitive ratio of 1 under perfect predictions), the algorithm lacks robustness: it can perform arbitrarily worse when predictions are inaccurate. This limitation has motivated a line of work on robustification methods, which can be broadly categorized into two families.

**Embedding-based methods** integrate prediction-driven logic directly into classical caching algorithms like MARKER [3]. For instance, PREDICTIVEMARKER [5] modifies the original eviction policy of MARKER based on predictions and achieves $4H_k$-robustness but not 1-consistency. Subsequently, LMARKER [6] improves smoothness and offers a $(2H_k + 4)$-robustness bound with 4-consistency. These algorithms derive their robustness from the marking mechanism inherent in MARKER, but this mechanism also restricts their ability to attain 1-consistency.

**Switching-based methods** switch between learning-augmented and classical algorithms based on past performance comparisons or the detection of prediction errors. For example, BLINDORACLE&LRU [7] switches between BLINDORACLE and LRU, following the currently better-performing one, and achieves $2k$-robustness and 2-consistency. More recently, F&R and F&R (FITF) [8] switch from a non-robust learning-augmented algorithm to a robust one upon detecting prediction errors, achieving $\mathcal{O}(\log k)$-robustness and 1-consistency. However, both F&R and F&R (FITF) have a total computational overhead of $\mathcal{O}(n^2 \log k)$, as they require recomputing the optimal solution over the entire observed request sequence upon each cache miss, with each recomputation costing $\mathcal{O}(n \log k)$.

We provide further related work in Appendix A, along with a comprehensive comparison of existing algorithms in Appendix B. Overall, while prior methods have made progress in improving robustness, they often do so at the cost of compromising 1-consistency or significantly increasing time complexity.

This leads to the central question of our work: *Can we enhance the robustness of learning-augmented caching algorithms in a time-efficient manner, without compromising 1-consistency?*

## 1.2 Our Contributions

**A New Framework.** We answer the above question affirmatively by introducing GUARD, a robustification framework applicable to any RB-following algorithm, which we formally define in Sec. 3.2, a broad class of learning-augmented caching algorithms, including BLINDORACLE. *Given an RB-following algorithm* A, GUARD&A *maintains the original asymptotic time complexity* [2] *of* A *while achieving the trade-off between consistency and robustness:* $(1, 2H_{k-1} + 2)$, *which is state-of-the-art.*

GUARD introduces a novel, lightweight phase-based mechanism for detecting prediction errors. In contrast to switching-based methods, it avoids recomputing costly optimal solutions online and monitoring the performance of alternative algorithms.

**New Algorithms and Empirical Evaluation.** We apply GUARD to three representative RB-following algorithms, i.e., BLINDORACLE [5], LRB [9], and PARROT [10], each using a different type of prediction. All resulting variants achieve 1-consistency and $(2H_{k-1} + 2)$-robustness, while maintaining their original asymptotic time complexities. Extensive experiments show that GUARD-based algorithms outperform existing methods and closely approach the state-of-the-art across a broad range of benchmarks with both synthetic and real-world predictors. The combination of strong theoretical guarantees and low runtime overhead underscores their value in practical applications.

---

[2]In this paper, we focus on the algorithm's time complexity, excluding the cost of predictor calls.

## 2 Preliminaries

The caching problem involves a universe of pages $\mathcal{P}$ and a cache of size $k$. A sequence of page requests $\sigma := r_1, r_2, ..., r_n$ must be processed online. Each request $r_i$ is a pair $(t_i, p_i)$, where $t_i$ denotes the time at which $p_i \in \mathcal{P}$ is requested. Upon receiving a request $r_i$, the algorithm must determine whether the corresponding page $p_i$ resides in the cache. If it does, the request results in a *cache hit*; otherwise, a *cache miss* occurs, and the page must be loaded into the cache. If the cache is full, an eviction decision must be made to accommodate the new page. The objective is to minimize the total number of cache misses over the entire sequence $\sigma$.

An *online algorithm* makes eviction decisions without knowledge of future requests, whereas an *offline algorithm* has complete knowledge of the entire request sequence in advance. For any algorithm A, let $A(\sigma)$ denote its cost when serving the sequence $\sigma$, defined as the number of cache misses incurred. In the case of randomized algorithms, we consider the expected cost. The performance of online algorithms is typically evaluated using the *competitive ratio*. An algorithm A has competitive ratio $\alpha$ if for every request sequence $\sigma$, the following holds:

$$A(\sigma) \leq \alpha \cdot \text{OPT}(\sigma) + c \tag{1}$$

where $\text{OPT}(\sigma)$ denotes the cost of an optimal offline algorithm, and $c$ is a constant independent of the input. In our algorithms, the additive constant $c$ will be zero. We also say that A is $\alpha$-competitive in this case. For brevity, we omit $\sigma$ when it is clear from the context.

For learning-augmented algorithms, the competitive ratio often takes the form $\min\{\gamma + f(\eta), \delta\}$, which encapsulates key performance metrics such as consistency, robustness, and smoothness. The parameter $\gamma$ reflects the performance when predictions are perfect (i.e., $\eta = 0$), making the algorithm $\gamma$-*consistent*. The bound $\delta$ ensures that the performance never degrades beyond this factor, no matter how inaccurate the predictions are, which defines $\delta$-*robustness*. The function $f(\eta)$ characterizes how performance degrades with increasing total prediction error $\eta$, reflecting the notion of $\mathcal{O}(f(\eta))$-*smoothness* when $\gamma$ is constant.

## 3 Algorithm Optimality and $1$-Consistency

To maintain 1-consistency during robustification, we begin by analyzing structural properties of optimal offline eviction policies and identifying a broad class of 1-consistent online learning-augmented algorithms. Let OPT denote the optimal offline algorithm that follows Belady's rule: *evicting the page whose next request occurs furthest in the future.* For each request $r_i = (t_i, p_i)$, let $T_i$ denote the time of the next request for page $p_i$ after $t_i$. Following Antoniadis et al. [11] and Song et al. [9], we define the *Belady's binary label* $y_i$ as follows:

$$y_i = \begin{cases} 1 & \text{if OPT evicts } p_i \text{ before } T_i, \\ 0 & \text{otherwise.} \end{cases} \tag{2}$$

We refer to the page $p_i$ as a *1-page* if $y_i = 1$, and a *0-page* if $y_i = 0$. Note that a page may be a 1-page at some times and a 0-page at others if it is loaded into the cache multiple times, depending on OPT's eviction decisions. By default, we assume that OPT serves the entire request sequence $\sigma$.

### 3.1 RB-Compliant Algorithms

Belady's binary labeling naturally gives rise to a class of optimal algorithms that, although not necessarily replicating the exact eviction decisions of OPT, still achieve optimal performance.

This is achieved through adherence to a principle we refer to as the relaxed Belady's rule: *prioritizing the eviction of 1-pages over 0-pages whenever a cache replacement is necessary.*

**Definition 3.1.** *An algorithm is said to be RB-compliant if it adheres to the relaxed Belady's rule.*

To analyze the behavior of such algorithms, we define the following notation. Let the sets $\mathbf{1}_i^A$ and $\mathbf{0}_i^A$ denote the sets of cached 1-pages and 0-pages, respectively, after algorithm A processes request $r_i$. Similarly, let $\mathbf{1}_i^*$ and $\mathbf{0}_i^*$ denote the corresponding sets for OPT. The following lemma provides a key structural property of RB-compliant algorithms: while their eviction decisions may differ from those of OPT, they maintain the same set of 0-pages in the cache at all times.

**Lemma 3.2.** *For an RB-compliant algorithm* A*, we have* $|\mathbf{1}_i^{\mathrm{A}}| = |\mathbf{1}_i^*|$ *and* $\mathbf{0}_i^{\mathrm{A}} = \mathbf{0}_i^*$ *after serving each request* $r_i$.

*Proof.* The proof is carried out by exhaustive case analysis. See Appendix C for details. □

This result implies that an RB-compliant algorithm always caches at least one 1-page when a cache miss occurs; otherwise, OPT would evict a 0-page at that time, contradicting the definition. Therefore, we have the following corollary:

**Corollary 3.3.** *RB-compliant algorithms evict only 1-pages.*

This ensures that RB-compliant algorithms match the performance of OPT exactly:

**Proposition 3.4.** *RB-compliant algorithms are optimal, incurring the same cost as* OPT.

*Proof.* Lemma 3.2 implies that an RB-compliant algorithm A maintains the same set of 0-pages as OPT at every step. Since 0-pages are precisely those pages that will be accessed again before any 1-page is needed, A incurs at least as many cache hits as OPT, and thus no more cache misses. Therefore, A is optimal. □

Proposition 3.5 reveals a key property concerning the ordering of next request times.

**Proposition 3.5.** *For an RB-compliant algorithm* A*, after serving each request* $r_i$*, the next request time of any page in* $\mathbf{1}_i^{\mathrm{A}}$ *is later than that of any page in* $\mathbf{0}_i^{\mathrm{A}}$.

*Proof.* The proof follows from several observations about the relationship between the pages cached by A and OPT. See Appendix D for details. □

## 3.2    RB-Following Algorithms

Building on the theory from the previous section, we now define a class of learning-augmented algorithms based on the relaxed Belady's rule. This class captures a wide range of learning-augmented algorithms that are 1-consistent.

**Definition 3.6.** *A learning-augmented algorithm is said to be RB-following if it prioritizes evicting 1-pages over 0-pages under perfect predictions.*

By definition, an RB-following algorithm is RB-compliant when predictions are accurate. As a result, all conclusions and performance guarantees derived for RB-compliant algorithms in Sec. 3.1 apply directly to RB-following algorithms under perfect predictions. Consequently, we have:

**Corollary 3.7.** *RB-following algorithms are* 1*-consistent.*

RB-following algorithms can leverage various types of predictions, including predicted next request times, Belady binary labels (as defined above), or even the optimal eviction choices made by OPT. Several representative learning-augmented algorithms are RB-following: BLINDORACLE [5] evicts the page with the furthest predicted next request time; LRB [9] randomly selects a predicted 1-page for eviction, prioritizing them over predicted 0-pages; PARROT [10] mimics the eviction decisions of OPT by evicting the page predicted to be requested furthest in the future (FitF). See Sec. 4.4 for further discussion of these algorithms. However, all the aforementioned RB-following algorithms follow predictions blindly, leading to a critical drawback: *poor robustness in the face of inaccurate predictions.* Lemma 3.8 shows that such algorithms can incur unbounded competitive ratios.

**Lemma 3.8.** *RB-following algorithms that blindly follow predictions have unbounded robustness.*

*Proof.* The proof presents a simple adversarial case. See Appendix E for details. □

This observation motivates *a robustification framework that preserves the 1-consistency of RB-following algorithms while significantly enhancing their robustness.*

# 4 GUARD: The Robustification Framework

## 4.1 Insights and Overview

We now introduce GUARD, a general framework that robustifies RB-following algorithms by mitigating the impact of prediction errors while preserving their 1-consistency under accurate predictions. The design of GUARD is grounded in three key observations as follows.

**Observation 1: Immediate protection hurts 1-consistency.** Marking-based learning-augmented caching algorithms, such as PREDICTIVEMARKER [5], LMARKER [6], and MARK&PREDICT [11], use a fixed mechanism to protect pages from eviction immediately after they are requested. While this method ensures bounded robustness, it may prevent necessary evictions that would otherwise be justified by accurate predictions. This observation motivates us to rethink the timing and conditions under which protection is applied, rather than applying it uniformly after every access. In particular, we aim to guard pages only when there is evidence of a prediction error, avoiding overprotection.

**Observation 2: Error detection must balance accuracy and efficiency.** Many switching-based methods detect prediction errors at runtime to switch between prediction-driven and robust fallback policies. A common method, as used by BLINDORACLE&LRU and BLINDORACLE&MARKER [7], involves comparing the current total cost incurred by the prediction-based algorithm with that of the fallback algorithm. However, this comparison is often insensitive: when the learning-augmented algorithm underperforms relative to the classical one, the accumulated prediction errors can be large, as the classical algorithm (e.g., LRU) may perform significantly worse than OPT. Consequently, this introduces a multiplicative factor of 2 in the robustness guarantees. Alternatively, some methods, such as F&R and F&R (FITF) [8], detect prediction errors by explicitly recomputing the optimal solution over the observed request sequence up to the point of a cache miss, and checking whether the optimal solution would also incur a miss. While this yields highly accurate error detection, its overhead is prohibitive for real-time use. This motivates an error detection mechanism that is lightweight, sensitive to errors that trigger cache misses, and practical for real systems.

**Observation 3: RB-compliant algorithms follow an intrinsic eviction pattern.** Any RB-compliant algorithm, including OPT, never retains an unrequested page in the cache during the time interval between the eviction and the next request of another page, revealing the underlying principle of optimal eviction decisions. We defer the proof to Appendix G. Formally, if a request $r_i$ results in a cache miss at time $t_i$, and the requested page $p_i$ was previously evicted at time $\mu < t_i$, then all pages remaining in the cache at time $t_i$ must have been requested at least once between $\mu$ and $t_i$. This behavior implies a kind of *causal chain* between evictions and requests, which motivates a way to identify mispredictions without recomputing OPT directly.

Inspired by the above, GUARD selectively guards (i.e., protects) a requested page $p_i$ from eviction only when a prediction error is detected, thereby preserving 1-consistency under accurate predictions. This mechanism ensures that pages that should be evicted before $p_i$ are indeed evicted within the same phase, maintaining bounded robustness even under poor predictions.

Algorithm 1 describes GUARD&A. Specifically, the execution of GUARD is divided into *phases*. At the start of each phase, all cached pages are labeled as old and stored in a set $\mathcal{U}$, which tracks unrequested old pages. A new phase begins when $\mathcal{U}$ becomes empty, indicating that all old pages have either been requested or evicted. The period from the beginning of execution until the first reset of $\mathcal{U}$ (Line 8) is referred to as *the 0-th phase*. When a cache miss occurs, if the requested page $p_i$ was previously evicted in the current phase (Line 10), this signals a potential prediction error. In response, GUARD evicts a random unguarded page from $\mathcal{U}$ and marks $p_i$ as guarded, preventing it from being evicted until the next phase. Otherwise, A's eviction policy is followed over the set of unguarded pages. This design leverages the intrinsic pattern of the optimal eviction behavior (Observation 3) to detect meaningful mispredictions efficiently without costly recomputation (Observation 2), while avoiding immediate protection (Observation 1) and insensitive performance comparisons (Observation 2).

We simplify A by omitting its specific eviction behavior and other auxiliary logic in Algorithm 1. This is because, in essence, GUARD operates as a companion process to A, *running concurrently, dynamically restricting the set of eviction candidates (Line 14-15) or overriding A's eviction policy (Line 11) when necessary.*

---
**Algorithm 1** GUARD&A
---
1: $\mathcal{U} \leftarrow \emptyset$ (the set tracks *unrequested old pages* in the cache)
2: **for** $i = 1, ..., n$ **do**
3:     Receive a page request $r_i = (t_i, p_i)$ at time $t_i$
4:     **if** $p_i$ *is not in the cache* **then**
5:         **if** *the cache is full* **then**
6:             **if** $\mathcal{U}$ is empty **then**
7:                 *Unguard all cached pages*
8:                 $\mathcal{U} \leftarrow \{$all cached pages$\}$ *(a new phase begins)*
9:             **end if**
10:             **if** $p_i$ *was evicted in the current phase* **then**
11:                 Evict a page $x$ from $\mathcal{U}$ uniformly at random
12:                 *Guard* $p_i$
13:             **else**
14:                 $\mathcal{S} \leftarrow \{$all unguarded cached pages$\}$
15:                 Follow A's policy to evict a page $x$ from $\mathcal{S}$ based on predictions
16:             **end if**
17:             **if** the evicted page $x \in \mathcal{U}$ **then**
18:                 $\mathcal{U} \leftarrow \mathcal{U} \backslash \{x\}$
19:             **end if**
20:         **end if**
21:         Load $p_i$ into the cache
22:     **end if**
23:     **if** $p_i \in \mathcal{U}$ **then**
24:         $\mathcal{U} \leftarrow \mathcal{U} \backslash \{p_i\}$
25:     **end if**
26: **end for**
---

## 4.2 Preserving 1-Consistency

**Proposition 4.1.** *Under perfect predictions, no page will be guarded by* GUARD&A.

*Proof.* Suppose, for contradiction, that a page $p_a$ is the first page ever guarded by GUARD&A, and it is guarded at time $t_i$ upon a request for $p_a$. Before $t_i$, no pages have been guarded, so GUARD&A behaves exactly like A up to this point. Consider the phase during which $p_a$ is guarded.

By the algorithm's logic, $p_a$ must have been evicted earlier in this phase, say, at time $\mu < t_i$. According to Corollary 3.3, $p_a$ is a 1-page at time $\mu$. At time $t_i$, since $p_a$ was previously evicted, GUARD&A enters the error-handling branch (Lines 10–12), implying there exists an unrequested old page $p_b \in \mathcal{U}$ at time $t_i$, whose next request occurs at time $T_b > t_i$. Then, $p_b$ was also a 1-page in the cache at time $\mu$, according to Proposition 3.5.

Now consider an alternative algorithm, GUARD&B, behaves identically to GUARD&A before $t_i$, except that it evicts $p_b$ instead of $p_a$ at time $\mu$, keeping $p_a$ in the cache until $t_i$. Because both $p_a$ and $p_b$ are 1-pages at time $\mu$, GUARD&B remains RB-compliant before $t_i$ under perfect predictions. At time $t_i$, GUARD&B would experience a cache hit for $p_a$, whereas OPT incurs a cache miss (since $p_a$ was a 1-page at time $\mu$ and was not accessed again until $t_i$). Moreover, GUARD&B and OPT incur the same number of cache misses before $t_i$ (see Proposition 3.4), implying GUARD&B performs better than OPT after serving requests up to time $t_i$, which contradicts the optimality of OPT.

Therefore, no such $p_a$ can exist, and no page is ever guarded under perfect predictions. $\qquad\square$

From Proposition 4.1, under perfect predictions, GUARD&A never guards any page and thus behaves identically to A. Since A is 1-consistent by definition, GUARD&A inherits the following property:

**Corollary 4.2.** *For any RB-following algorithm* A, GUARD&A *is 1-consistent.*

### 4.3 Enhancing Robustness

Our preliminary proof shows the $\mathcal{O}(\log k)$-robustness of GUARD&A. A more refined analysis (see Appendix J) tightens this bound to $2H_{k-1} + 2$.

To establish the robustness bound, we introduce several key notations. Let the final phase executed by GUARD&A be the $Q$-th phase and denote by $\mathcal{Q}$ the set $\{0, 1, ..., Q\}$. A page request $r_i$ is *distinct* within a phase if the requested page $p_i$ has not been previously requested during that phase. Let $c_q$ denote the number of distinct new pages requested during the $q$-th phase.

Among the requests leading to evictions in the $q$-th phase, let $n_q$ denote the number of requests for new pages and $o_q$ denote the number of requests for old pages. We decompose $n_q$ into $n_q^{new}$, representing the number of requests that cause the eviction of new pages, and $n_q^{old}$, representing the number of requests that result in the eviction of old pages. Therefore, we have $n_q = n_q^{new} + n_q^{old}$. This notation allows us to precisely track the types of evictions occurring during each phase.

The following lemma gives a lower bound on the cost of OPT on sequence $\sigma$. Note that, in this paper, we denote OPT$(\sigma)$ by OPT for brevity.

**Lemma 4.3.** $\sum\limits_{q \in \mathcal{Q}} \frac{1}{2} c_q \leq \text{OPT} \leq \sum\limits_{q \in \mathcal{Q}} n_q^{old}.$

*Proof.* The proof builds on a phase-based analysis, following the approach of Fiat et al. [3]. See Appendix I for details. $\square$

**Lemma 4.4.** *For each $q \in \mathcal{Q}$, $n_q \leq 2c_q$ and $n_q^{old} \leq c_q$.*

*Proof.* Each distinct new page can be loaded into the cache at most twice per phase, as it becomes guarded upon its second request. Hence, $n_q \leq 2c_q$.

If a new page is loaded into the cache twice within a phase, it must be evicted by a request for another new page before its second request. This is because, if a request for an old page results in a cache miss, GUARD&A only evicts an old page from $\mathcal{U}$. Thus, we have $n_q - c_q \leq n_q^{new}$, which implies $n_q^{old} = n_q - n_q^{new} \leq c_q$. $\square$

**Theorem 4.5.** GUARD&A *is $\mathcal{O}(\log k)$-robust.*

*Proof.* We bound the number of evictions. In phase $q$, let $n_q$ be the number of cache misses due to new pages, and $o_q$ those due to old pages. From Lemma 4.4, $n_q \leq 2c_q$. To upper bound $o_q$, we make the following assumptions, each of which can only increase the number of cache misses.

1. The number of distinct old-page requests is $k$ (the maximum possible).

2. All $n_q^{old}$ evictions happen before requests for old pages.

3. Evicted old pages have earlier next request times than remaining ones.

On each cache miss for an old-page request, an unrequested page from $\mathcal{U}$ is evicted and the requested page is guarded. Let $p_j$ denote the probability of a cache miss for the $j$-th subsequent distinct old page request. We have

$$p_j = \begin{cases} 1, & \text{if } 1 \leq j \leq n_q^{old}, \\ \frac{n_q^{old}}{k-(j-1)}, & \text{if } j > n_q^{old}. \end{cases} \tag{3}$$

The expected value of $o_q$ is bounded as follows:

$$\mathbb{E}[o_q] \le n_q^{old} + \sum_{j=n_q^{old}+1}^{k} \min\left\{p_j, 1\right\}$$

$$= 2n_q^{old} + \sum_{j=n_q^{old}+1}^{k-n_q^{old}} \frac{n_q^{old}}{k-(j-1)}$$

$$= \left(2 + H_{k-n_q^{old}} - H_{n_q^{old}}\right)n_q^{old}$$

$$\le \left(H_k + 1\right)c_q. \quad \text{(by Lemma 4.4)} \tag{4}$$

The inequalities hold regardless of whether $q < Q$ or $q = Q$. Let $\text{GA}_q$ denote the number of cache misses incurred by GUARD&A during the $q$-th phase. By (4), GUARD&A's total expected cost $\mathbb{E}[\text{GA}]$ is bounded by:

$$\mathbb{E}[\text{GA}] = c_0 + \sum_{q \in \mathcal{Q}} \mathbb{E}[\text{GA}_q] = c_0 + \sum_{q \in \mathcal{Q}} \left(n_q + \mathbb{E}[o_q]\right)$$

$$\le (H_k + 3)\sum_{q \in \mathcal{Q}} c_q \le (2H_k + 6)\text{OPT}. \quad \text{(by Lemmas 4.3 and 4.4)} \tag{5}$$

This completes the proof. $\qquad\square$

The above assumptions simplify the proof and immediately yield logarithmic robustness, but inevitably loosen the result. Theorem 4.6 presents a tight bound.

**Theorem 4.6.** GUARD&A *is* $(2H_{k-1} + 2)$*-robust, which is tight.*

*Proof.* The proof analyzes eviction chains that account for each eviction, without relying on any assumptions. See Appendix J for details. $\qquad\square$

### 4.4 Applications

We apply GUARD to three RB-following algorithms: BLINDORACLE [5], LRB [9], and PARROT [10]. These algorithms are selected because they are representative and rely on predictors that are practical in implementation. However, as they blindly follow predictions, they are 1-consistent but not robust. After being robustified via GUARD, each algorithm gains $(2H_{k-1} + 2)$-robustness while retaining 1-consistency. Implementing GUARD incurs an $\mathcal{O}(1)$ overhead per request when using hash tables, so the time complexity remains asymptotically the same as that of the base algorithm.

BLINDORACLE [5] evicts the page with the furthest predicted next request time (NRT), as detailed in Algorithm 3 in Appendix F. LRB [9] is an algorithm that has been applied to content distribution network caching. See Algorithm 4. It predicts Belady's binary labels and prioritizes eviction of predicted 1-pages over predicted 0-pages, labeled by the OPT that starts from the current cache content of LRB and serves subsequent requests in $\sigma$. Note that LRB remains RB-following, as proven in Appendix H. PARROT [10] learns to imitate the optimal policy using a neural network model. See Algorithm 5 for details. It directly evicts the predicted FitF (furthest-in-the-future) page, where FitF follows the definition in Sadek and Elias [8]. GUARD is also applicable to more sophisticated RB-following algorithms that do not blindly follow predictions, which we leave as future work.

Table 1: Robustified RB-following algorithms. B.O. stands for BLINDORACLE. NRT refers to the next request time, while FitF stands for furthest-in-the-future.

| Algorithm | Prediction | Consistency | Robustness | Smoothness | Time Complexity |
|---|---|---|---|---|---|
| GUARD&B.O. | NRT | 1 | $2H_{k-1} + 2$ | $\mathcal{O}(\log(\eta_t/\text{OPT}))$ | $\mathcal{O}(n \log k)$ |
| GUARD&LRB | Binary | 1 | $2H_{k-1} + 2$ | $\mathcal{O}(H_k \cdot \eta_b/\text{OPT})$ | $\mathcal{O}(n)$ |
| GUARD&PARROT | FitF Page | 1 | $2H_{k-1} + 2$ | $\mathcal{O}(H_k \cdot \eta_f/\text{OPT})$ | $\mathcal{O}(n)$ |

Refer to Table 1 for a summary, and to Appendix K for their pseudo-codes, smoothness proofs, and implementation details. $\eta_t$, $\eta_b$, and $\eta_f$ are error measures for different types of predictions, representing the total $\ell_1$ error of predicted next request times, the number of incorrect predictions of Belady's binary labels, and the number of incorrect predictions of FitF pages, respectively. A comprehensive comparison of existing algorithms is provided in Table 4 in Appendix B.

### 4.5 Achieving Better Smoothness with EXGUARD

Inspired by F&R [8] and ADAPTIVEQUERY-B [12], we explore how to trade off smoothness against predictor usage. The challenge lies in maintaining logarithmic robustness while adjusting the use of predictions. We propose an extension of GUARD, called EXGUARD, which allows for improved smoothness at the cost of increased predictor usage. EXGUARD constrains subsequent random evictions triggered by a misprediction-induced eviction, thereby avoiding excessive conservativeness and improving smoothness. Meanwhile, EXGUARD maintains 1-consistency and $\mathcal{O}(\log k)$-robustness.

The learning-augmented algorithms in this paper invoke the predictor upon eviction, following common implementations in prior caching systems [9, 13, 14]. GUARD&B.O. and GUARD&PARROT invoke the predictor $\mathcal{O}(\mathrm{OPT})$ times, whereas EXGUARD&B.O. and EXGUARD&PARROT use the predictor $\mathcal{O}(d \cdot \mathrm{OPT})$ times, where $d \in [1, H_k]$. EXGUARD&BLINDORACLE achieves $\mathcal{O}\big(\min(\log(\eta_t/\mathrm{OPT}), \lambda\sqrt{\eta_t/\mathrm{OPT}})\big)$-smoothness, where $\lambda = h/e^h \le 1/e$. Here, $h = H_k/(2d)$, and thus $h \in [1/2, H_k/2]$. EXGUARD&PARROT achieves $\mathcal{O}(H_k/d \cdot \eta_f/\mathrm{OPT})$-smoothness. Algorithms using EXGUARD are also included in Table 4 in the Appendix for comparison. In Appendix N, we present a comprehensive introduction to EXGUARD, including its empirical results and formal proofs of the favorable trade-offs it achieves.

## 5 Experiments

We now present a comprehensive evaluation of learning-augmented caching algorithms to assess their performance under both synthetic and real-world predictions.

### 5.1 Experimental Setup

Building on the experimental frameworks of Liu et al. [10] and Chłędowski et al. [15], we construct an expanded benchmark that includes more algorithm variants, datasets, and prediction types. We evaluate algorithms using three types of predictions: *Next request times (NRT)*, *Belady's binary labels*, and *FitF page predictions*. We also include F&R [8], whose action predictions can be derived from predicted next request times as demonstrated in Sadek and Elias [8]. For switching-based algorithms, we follow Chłędowski et al. [15], setting a deterministic switching bound of 1 and a randomized weight $\beta = 0.99$. Some algorithms are omitted due to either performance equivalence or excessive computational overhead.

**Datasets.** We use BrightKite [16] and Citi [17], with cache sizes set to 10 and 100, respectively, following Lykouris and Vassilvtiskii [5]. We further use SPEC CPU2006 memory traces [18] to evaluate real-world performance. Following Chłędowski et al. [15] and Liu et al. [10], we adopt the 16-way 2MB cache configuration for consistency.

**Predictions.** We consider both synthetic and real predictors. For synthetic predictions, NRT and binary label predictions are processed differently. To simulate NRT predictions, we add log-normal noise to true request times. Pages without future requests are assigned a value of $n + 1$, where $n$ is the number of requests. On the other hand, binary label predictions are generated by flipping true Belady labels with a given probability to simulate noisy prediction scenarios. For real-world predictors, we consider the following: (i) PLECO [19]: A probability-based predictor that estimates the access likelihood $p$ of a page and predicts its next request after $1/p$ steps; (ii) POPU [20]: A frequency-based predictor that assumes a page requested in fraction $p$ of past accesses will reappear after $1/p$ steps; (iii) LRB Predictor: Uses LightGBM [21] to predict next request times, then classifies pages beyond the Belady boundary as predicted 1-pages and others as predicted 0-pages, consistent with Song et al. [9].

## 5.2 Experimental Results

All cost ratios are reported relative to OPT. Figures 1 and 2 show algorithm performance under varying levels of synthetic next request time (NRT) and binary prediction errors on the BrightKite dataset. Below, we abbreviate BLINDORACLE (B.O.), LMARKER (LM.), LNONMARKER (LNONM.), PREDICTIVEMARKER (P.M.), and MARK&PREDICT (M.&P.). Due to differing interpretations of binary labels, only synthetic results for M.&P. are included. Both GUARD&B.O. and GUARD&LRB exhibit empirical 1-consistency and bounded robustness, aligning with theoretical guarantees.

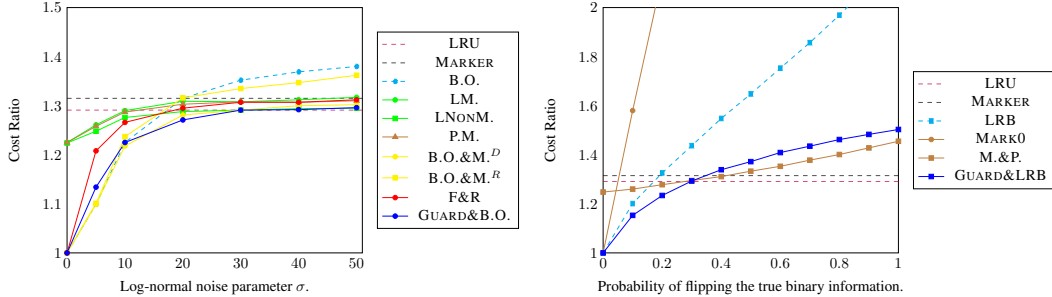

Figure 1: Performance with synthetic predictions of NRT on BrightKite.

Figure 2: Performance with synthetic predictions of binary labels on BrightKite.

Table 2: Average cost ratios on SPEC CPU2006 using PLECO and POPU predictors.

| Predictor | LRU | B.O. | LM. | LNONM. | P.M. | B.O.&M.$^D$ | F&R | GUARD&B.O. |
|---|---|---|---|---|---|---|---|---|
| PLECO | 1.478 | 1.404 | 1.335 | 1.346 | 1.335 | 1.294 | 1.360 | **1.226** |
| POPU | 1.478 | 1.261 | 1.320 | 1.312 | 1.312 | 1.233 | 1.319 | **1.203** |

Table 3: Average cost ratios on SPEC CPU2006 Benchmark using LRB predictor.

| Predictor | LRU | MARKER | LRB | MARK0 | GUARD&LRB |
|---|---|---|---|---|---|
| LRB predictor | 1.478 | 1.394 | 1.281 | 1.268 | **1.171** |

We further evaluate the algorithms on the SPEC CPU2006 benchmark using real-world predictors. Average cost ratios across all 13 datasets are shown in Tables 2 and 3. B.O.&M.$^R$ is omitted due to inferior performance compared to B.O.&M.$^D$. Results show that both GUARD&B.O. and GUARD&LRB achieve the lowest average cost ratios when using their respective predictors.

Additional results, including per-dataset performance and those using the "FitF page" predictor, are presented in Appendix M. We implement a new benchmark, Cache-Coliseum, for comprehensive comparison of learning-augmented algorithms (including ours), which is publicly available at `https://github.com/OptiSys-ZJU/cache-coliseum`.

## 6 Conclusion

In this paper, we introduced GUARD, a framework designed to robustify learning-augmented caching algorithms that follow the relaxed Belady's rule. GUARD preserves 1-consistency, improves robustness to $2H_{k-1} + 2$, and incurs minimal additional computation, making it highly practical for real-world applications. Experiments across multiple datasets and predictors show that GUARD&B.O., GUARD&LRB, and GUARD&PARROT achieve the best or near-best performance. These results validate the effectiveness of GUARD in enhancing both theoretical guarantees and empirical performance.

Our results highlight the potential of lightweight robustification techniques for integrating machine learning to enhance existing caching systems. Future work may explore: (1) whether robustness and smoothness can be further improved toward their respective theoretical lower bounds, as discussed in Appendix L, while still preserving 1-consistency and the asymptotic time complexity of the base algorithm; and (2) exploring robust learning-augmented algorithms under other caching models that arise in modern systems, as discussed in Appendix O, where, for example, the conventional constraint that "the requested item is always stored" is relaxed.

## Acknowledgments and Disclosure of Funding

This work was supported in part by the National Key Research and Development Program of China under Grant 2022YFB4500100, the National Science Foundation of China (62125206, 62502441), the Major Program of the National Natural Science Foundation of Zhejiang (LD25F020002), and the Singapore Ministry of Education under Academic Research Fund Tier 2 Award MOE-T2EP20122-0007 and Tier 1 Award RG23/23. Hailiang Zhao's work was supported in part by the Zhejiang University Education Foundation Qizhen Scholar Foundation.

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

# Appendix

## A   Further Related Work

**Learning-Augmented Algorithms.** The concept of learning-augmented algorithms was pioneered by Lykouris and Vassilvtiskii [5], who introduced the idea of integrating machine-learned predictions into online decision-making frameworks. Since then, this paradigm has gained traction across a wide range of algorithmic domains, including index structures [22], ski rental problems [23, 24], the Bahncard problem [25, 26], online knapsack [27, 28], online TSP [29, 30], $k$-server problems [31], metrical task systems (MTS) [20, 32], secretary problems [33, 34], graph-related tasks [35, 36], and data structures [37–39]. These techniques have also found applications in broader system-level contexts such as networking [9, 40] and caching [41, 42].

**Learning-Augmented Caching.** Recently, there has been growing interest in applying learning-augmented methods to the caching problem. The first robust learning-augmented caching algorithm, PREDICTIVEMARKER, was proposed by Lykouris and Vassilvtiskii [5]. It achieves a competitive ratio of $\min\{2 + 2\sqrt{4\eta_t/\text{OPT} + 1}, 4H_k\}$, where OPT is the optimal offline cost and $\eta_t$ represents the $l_1$-error of the prediction on next request times. This bound was later improved by Rohatgi [6], who introduced LMARKER, achieving a smoothness guarantee of $\mathcal{O}(\log(\eta_t/\text{OPT}))$. The same work also proposed LNONMARKER, which offers superior smoothness but sacrifices bounded robustness. Wei [7] further improved the competitive ratio by proposing a simple approach to combine BLINDORACLE with LRU, and proved its competitive ratio to be $2\min\{1 + 2\eta_t/\text{OPT}, 4 + 4/(k-1) \cdot \eta_t/\text{OPT}, k\}$. Wei [7] also combined BLINDORACLE with the randomized algorithm EQUITABLE, and proved that its competitive ratio is $(1 + \gamma)\min\{1 + 2\eta_t/\text{OPT}, 4 + 4/(k-1) \cdot \eta_t/\text{OPT}, H_k\}$, plus an additive constant $\mathcal{O}(k/\gamma)$, where $\gamma \in (0, 1/4)$ is a trade-off parameter.

Antoniadis et al. [20] studied the succinctness of the prediction. They show that it suffices to receive predictions of size $\mathcal{O}(\log k)$ per request, indicating which page should be evicted. This was further simplified by Antoniadis et al. [11], who demonstrated that binary predictions indicating whether a cached page is expected to be evicted can still yield strong performance. This practical insight inspired our exploration of caching with binary predictions and led to the development of the GUARD framework based on this minimalistic form of side information.

Im et al. [12] and Sadek and Elias [8] approached the problem differently, focusing on limiting the number of times predictions are used. Their work highlights the trade-off between the frequency of prediction queries and overall algorithmic performance, which is a consideration of particular relevance in settings where acquiring predictions is costly or resource-intensive.

## B   Comparative Analysis of Existing Algorithms

Table 4 provides a comprehensive comparison of recent learning-augmented caching algorithms, focusing on their performance guarantees and prediction requirements. Below, we analyze key algorithmic paradigms and design principles that distinguish these approaches.

**Prediction Types.** The literature explores various forms of predictions, each offering different levels of information and utility.

1. *Next Request Time Prediction.* The most commonly used prediction type is the next request time of a page, which is utilized by many existing algorithms, including BLINDORACLE [5], PREDICTIVEMARKER [5], LMARKER [6], and LNONMARKER [6].

2. *Binary Prediction.* Binary predictions convey specific binary information and have been explored both in systems [41, 42] and theoretical studies [11]. For example, MARK0 predicts Belady's binary labels to guide eviction decisions. MARK&PREDICT predicts whether a page will be requested during a given phase, rather than estimating the exact request time.

3. *FitF (Furthest-in-the-Future) Page Prediction.* Given the cache content, the predictor directly identifies the cached page that will be requested furthest in the future. It is used by F&R (FitF) [8], and PARROT [9].

Table 4: **Comparison of Learning-Augmented Caching Algorithms. Prediction type** categorizes the algorithms listed in the first column, including NRT (Next Request Time), Binary information (e.g., Belady's binary label or whether a page will be accessed in the current phase), FitF (Furthest-in-the-Future page to be requested), and Action (e.g., cache content of an optimal offline algorithm). **Abbreviations** include PREDICTIVEMARKER (P.M.), LNON-MARKER (LNonM.), LMARKER (LM.), BLINDORACLE (B.O.), EQUITABLE (EQ.), BLINDORA-CLE&MARKER (B.O.&M.), MARK&PREDICT (M.&P.), and PARROT (PA.). The superscripts $D$ and $R$ denote deterministic switching and randomized switching, respectively. For B.O.&M.$^R$ and B.O.&EQ.$^R$, $\gamma \in (0, \frac{1}{4})$ is a tunable trade-off parameter. **Time.** refers to the time complexity of the algorithm, excluding the cost of making predictor calls. $n$ is the total number of page requests. The $\log k$ in the time complexity of algorithms using NRT predictions comes from selecting the page with the furthest next request time. **Cons.** denotes consistency (i.e., the competitive ratio when predictions are perfect). **Coefficients and constants** presented in robustness are mostly obtained directly from the conclusions of the respective papers, while those for F&R and F&R (FITF) are derived from inequalities and may not be tight. **Error measures** $\eta_t$, $\eta_b$, $\eta_a$, and $\eta_f$ use different units across different prediction types, so their magnitudes are not directly comparable. For EXGUARD, $d \in [1, H_k]$. For EXGUARD&B.O., $\lambda = h/e^h \le 1/e$, where $h = H_k/(2d)$, and thus $h \in [1/2, H_k/2]$. For F&R (FITF), parameter $b \in \{1, ..., \log k\}$.

| NRT | Time. | Cons. | Robustness | Smoothness |
|---|---|---|---|---|
| P.M. [5] | $\mathcal{O}(n \log k)$ | 2 | $4H_k$ | $\mathcal{O}(\sqrt{\eta_t/\text{OPT}})$ |
| LM. [6] | $\mathcal{O}(n \log k)$ | 4 | $2H_k + 4$ | $\mathcal{O}(\log(\eta_t/\text{OPT}))$ |
| LNonM. [6] | $\mathcal{O}(n \log k)$ | 4 | $\infty$ | $\mathcal{O}(\log(k)/k \cdot \eta_t/\text{OPT})$ |
| B.O. [5] | $\mathcal{O}(n \log k)$ | 1 | $\infty$ | $\mathcal{O}(1/k \cdot \eta_t/\text{OPT})$ |
| B.O.&LRU [7] | $\mathcal{O}(n \log k)$ | 2 | $2k$ | $\mathcal{O}(1/k \cdot \eta_t/\text{OPT})$ |
| B.O.&M.$^D$ [7] | $\mathcal{O}(n \log k)$ | 2 | $4H_k - 2$ | $\mathcal{O}(1/k \cdot \eta_t/\text{OPT})$ |
| B.O.&M.$^R$ [7] | $\mathcal{O}(n \log k)$ | $1 + \gamma$ | $(1 + \gamma)(2H_k - 1) + \mathcal{O}(\frac{k}{\gamma})$ | $\mathcal{O}(1/k \cdot \eta_t/\text{OPT})$ |
| B.O.&EQ.$^R$ [7] | $\mathcal{O}(nk^2)$ | $1 + \gamma$ | $(1 + \gamma)H_k + \mathcal{O}(\frac{k}{\gamma})$ | $\mathcal{O}(1/k \cdot \eta_t/\text{OPT})$ |
| GUARD&B.O. | $\mathcal{O}(n \log k)$ | 1 | $2H_{k-1} + 2$ | $\mathcal{O}(\log(\eta_t/\text{OPT}))$ |
| ExGUARD&B.O. | $\mathcal{O}(n \log k)$ | 1 | $2H_{k-1} + \mathcal{O}(d)$ | $\mathcal{O}(\min(\log(\frac{\eta_t}{\text{OPT}}), \lambda\sqrt{\frac{\eta_t}{\text{OPT}}}))$ |

| Binary | Time. | Cons. | Robustness | Smoothness |
|---|---|---|---|---|
| M.&P. [11] | $\mathcal{O}(n)$ | 2 | $4H_k + \mathcal{O}(1)$ | $\mathcal{O}(H_k \cdot \eta_b/\text{OPT})$ |
| MARK0 [11] | $\mathcal{O}(n)$ | 1 | $\infty$ | $\mathcal{O}(H_k \cdot \eta_b/\text{OPT})$ |
| GUARD&LRB | $\mathcal{O}(n)$ | 1 | $2H_{k-1} + 2$ | $\mathcal{O}(H_k \cdot \eta_b/\text{OPT})$ |
| ExGUARD&LRB | $\mathcal{O}(n)$ | 1 | $2H_{k-1} + \mathcal{O}(d)$ | $\mathcal{O}(H_k/d \cdot \eta_b/\text{OPT})$ |

| Action | Time. | Cons. | Robustness | Smoothness |
|---|---|---|---|---|
| F&R [8] | $\mathcal{O}(n^2 \log k)$ | 1 | $3(1 + \log k) + \mathcal{O}(1)$ | $\mathcal{O}(\log(\eta_a/\text{OPT}))$ |

| FitF | Time. | Cons. | Robustness | Smoothness |
|---|---|---|---|---|
| F&R (FITF) [8] | $\mathcal{O}(n^2 \log k)$ | 1 | $9(1 + 1/b) \log k + \mathcal{O}(b)$ | $\mathcal{O}(\log(k)/b \cdot \eta_f/\text{OPT})$ |
| GUARD&PA. | $\mathcal{O}(n)$ | 1 | $2H_{k-1} + 2$ | $\mathcal{O}(H_k \cdot \eta_f/\text{OPT})$ |
| ExGUARD&PA. | $\mathcal{O}(n)$ | 1 | $2H_{k-1} + \mathcal{O}(d)$ | $\mathcal{O}(H_k/d \cdot \eta_f/\text{OPT})$ |

4. *Action Prediction.* Introduced by Antoniadis et al. [20], action predictions have been primarily used in metrical task systems (MTS). In caching, they indicate the cache contents of an optimal algorithm. F&R [8] is a recent example that leverages this type of prediction.

**Design Principles.** Below we compare the underlying designs of existing algorithms in detail.

- Embedding-based methods integrate additional algorithmic logic directly into existing caching algorithms to refine eviction strategies. For instance, PREDICTIVEMARKER [5] and LMARKER [6] refine eviction policies by leveraging eviction chains, i.e., a sequence of evictions where each eviction is triggered by a cache miss caused by a prior eviction. MARK&PREDICT [11] augments the MARKER algorithm by incorporating binary predictions to prioritize certain pages for eviction. The MARKER framework marks a requested page and prohibits its eviction until the end of a phase. While embedding into MARKER

enables the aforementioned algorithms to achieve logarithmic robustness, this approach inherently limits the best achievable consistency to 2. This limitation arises because the MARKER framework prevents evicting some marked pages, even when OPT would prioritize their eviction.

Instead, some algorithms adopt high-level principles from the MARKER framework without strictly adhering to its structure. For example, LNONMARKER [6] and MARK0 [11] modify the definition of phases. They both achieve 1-consistency but have unbounded robustness.

- Switching-based methods dynamically alternate between algorithms based on historical performance or detected prediction errors. This approach was first introduced in the seminal work of Fiat et al. [3] and later refined by Blum and Burch [43]. The algorithms proposed by Wei [7] switch between a 1-consistent learning-augmented algorithm and a classical robust algorithm. Their deterministic variant achieves a consistency of 2, while the randomized version attains $(1 + \gamma)$-consistency for $\gamma \in (0, 1/4)$. Although $(1 + \gamma)$ appears promising, the competitive ratio of the randomized version includes a non-negligible constant term $\mathcal{O}(k/\gamma)$, which degrades practical performance. Sadek and Elias [8] further refine switching-based methods by adopting a more fine-grained approach. Unlike Wei [7], F&R [8] switches to a robust algorithm only when prediction errors are detected on the fly, thereby ensuring 1-consistency. However, the detection mechanism is overly sensitive, resulting in high algorithmic complexity of $\mathcal{O}(n^2 \log k)$. The key challenge of switching-based algorithms lies in determining the switching timing.

## C  Proof of Lemma 3.2

**Lemma 3.2.** *For any RB-compliant algorithm* A*, we have* $|\mathbf{1}_i^A| = |\mathbf{1}_i^*|$ *and* $\mathbf{0}_i^A = \mathbf{0}_i^*$ *after serving each request* $r_i$.

*Proof.* The proof proceeds by induction. Suppose the cache becomes full for the first time after serving $m$ page requests. For any $i = 1, ..., m$, the lemma holds trivially, as no eviction occurs for either A or OPT. Now assume that at time $t_i$ ($i \geq m$), the conditions $|\mathbf{1}_i^A| = |\mathbf{1}_i^*|$ and $\mathbf{0}_i^A = \mathbf{0}_i^*$ hold. Consider the following two cases when $r_{i+1}$ arrives at time $t_{i+1}$:

**Case (i)**  $p_{i+1} \in \mathbf{0}_i^*$. By hypothesis $\mathbf{0}_i^A = \mathbf{0}_i^*$, we have $p_{i+1} \in \mathbf{0}_i^A$. Since $\mathbf{0}_i^*$ and $\mathbf{0}_i^A$ are the sets of cached 0-pages for OPT and A, respectively, $r_{i+1}$ results in a cache hit for both A and OPT.

   (a) If $y_{i+1} = 1$, $p_{i+1}$ is a 1-page at this moment. Then $\mathbf{1}_{i+1}^* = \mathbf{1}_i^* \cup \{p_{i+1}\}$ and $\mathbf{1}_{i+1}^A = \mathbf{1}_i^A \cup \{p_{i+1}\}$. Since $|\mathbf{1}_i^*| = |\mathbf{1}_i^A|$, it follows that $|\mathbf{1}_{i+1}^*| = |\mathbf{1}_{i+1}^A|$. On the other hand, $\mathbf{0}_{i+1}^* = \mathbf{0}_i^* \setminus \{p_{i+1}\} = \mathbf{0}_i^A \setminus \{p_{i+1}\} = \mathbf{0}_{i+1}^A$.

   (b) If $y_{i+1} = 0$, $p_{i+1}$ is a 0-page at this moment. Then $\mathbf{1}_{i+1}^* = \mathbf{1}_i^*$ and $\mathbf{1}_i^A = \mathbf{1}_{i+1}^A$. Thus, $|\mathbf{1}_{i+1}^*| = |\mathbf{1}_i^*| = |\mathbf{1}_i^A| = |\mathbf{1}_{i+1}^A|$. Similarly, $\mathbf{0}_{i+1}^* = \mathbf{0}_i^* = \mathbf{0}_i^A = \mathbf{0}_{i+1}^A$.

**Case (ii)**  $p_{i+1} \notin \mathbf{0}_i^*$. Note that $p_{i+1} \notin \mathbf{1}_i^*$ must hold because OPT evicts 1-pages at some time before they are requested again. Since $p_{i+1}$ is neither in $\mathbf{0}_i^*$ nor in $\mathbf{1}_i^*$, $r_{i+1}$ results in a cache miss for OPT. By Belady's rule, OPT evicts the page with the furthest next request time, denoted as $p_x$. This page $p_x$ must be a 1-page by definition, implying $|\mathbf{1}_i^*| > 0$.

   Then we consider A. By hypothesis $\mathbf{0}_i^A = \mathbf{0}_i^*$, we have $p_{i+1} \notin \mathbf{0}_i^A$. By hypothesis $|\mathbf{1}_i^A| = |\mathbf{1}_i^*|$, we have $|\mathbf{1}_i^A| > 0$. Therefore, if $r_{i+1}$ results in a cache miss for A, it evicts a 1-page from $|\mathbf{1}_i^A|$, denoted as $p_y$; otherwise, $r_{i+1}$ results in a cache hit for A, indicating $p_{i+1} \in \mathbf{1}_i^A$.

   (a) If $y_{i+1} = 1$, $p_{i+1}$ is a 1-page at this moment. Then $\mathbf{1}_{i+1}^* = \mathbf{1}_i^* \setminus \{p_x\} \cup \{p_{i+1}\}$. For A, $|\mathbf{1}_{i+1}^A| = |\mathbf{1}_i^A|$ regardless of whether $r_{i+1}$ results in a cache hit or not for A. Thus, $|\mathbf{1}_{i+1}^*| = |\mathbf{1}_{i+1}^A|$. Moreover, $\mathbf{0}_{i+1}^* = \mathbf{0}_i^* = \mathbf{0}_i^A = \mathbf{0}_{i+1}^A$.

   (b) If $y_{i+1} = 0$, $p_{i+1}$ is a 0-page at this moment. Then $\mathbf{1}_{i+1}^* = \mathbf{1}_i^* \setminus \{p_x\}$ and $\mathbf{0}_{i+1}^* = \mathbf{0}_i^* \cup \{p_{i+1}\}$. For A, if $r_{i+1}$ results in a cache miss, $\mathbf{1}_{i+1}^A = \mathbf{1}_i^A \setminus \{p_y\}$ and $\mathbf{0}_{i+1}^A = \mathbf{0}_i^A \cup \{p_{i+1}\}$; if $r_{i+1}$ results in a cache hit, $\mathbf{1}_{i+1}^A = \mathbf{1}_i^A \setminus \{p_{i+1}\}$ and $\mathbf{0}_{i+1}^A = \mathbf{0}_i^A \cup \{p_{i+1}\}$. Thus, $|\mathbf{1}_{i+1}^A| = |\mathbf{1}_{i+1}^*|$ and $\mathbf{0}_{i+1}^A = \mathbf{0}_{i+1}^*$ always hold.

In both cases, $|\mathbf{1}_{i+1}^{A}| = |\mathbf{1}_{i+1}^{*}|$ and $\mathbf{0}_{i+1}^{A} = \mathbf{0}_{i+1}^{*}$, completing the induction. Additionally, we observe that A experiences no more cache misses than OPT from the above analysis. □

# D Proof of Proposition 3.5

We first establish three lemmas to facilitate the proof of Proposition 3.5.

**Lemma D.1.** *For OPT, after serving each request $r_i$, the next request time of any page in $\mathbf{1}_i^{*}$ is later than that of any page in $\mathbf{0}_i^{*}$.*

*Proof.* Suppose that there exists a pair of 1-page $a_1$ and 0-page $a_0$ in the OPT's cache such that the next arrival time of $a_1$ is smaller than that of $a_0$. That is to say, $h_1 < h_0$. Assume $a_1$ is evicted at time $\mu_1$, then $\mu_1 < h_1$ since a 1-page is evicted before it is requested again. $a_0$ is also cached at time $\mu_1$ since $\mu_1 < h_1 < h_0$, and it will be kept cached before $h_0$ by the definition of 0-page. This implies at the time $\mu_1$, OPT evicts a page whose next arrival time is not the largest, which contradicts the definition. □

**Lemma D.2.** *For any algorithm A, after serving each request $r_i$, $\mathbf{0}_i^{A} \subseteq \mathbf{0}_i^{*}$ holds.*

*Proof.* Consider any 0-page $p_j \in \mathbf{0}_i^{A}$ at some time $\mu \in (t_i, t_{i+1})$. Its last request occurs at time $t_j \leq t_i < \mu$, and its next request occurs at time $T_j > \mu$. Immediately after its last request at $t_j$, $p_j$ must be in OPT's cache, regardless of whether OPT experienced a cache hit or not at $t_j$. Since $p_j$ is a 0-page, OPT will not evict it before $T_j$. Thus, at time $\mu$, we have $p_j \in \mathbf{0}_i^{*}$, which proves the lemma. □

For OPT and an RB-compliant algorithm A, let $\mathbf{H}_i^{*}$ and $\mathbf{H}_i^{A}$ denote the *ordered sets* of the next request times of 1-pages in $\mathbf{1}_i^{*}$ and $\mathbf{1}_i^{A}$ after serving request $r_i$, respectively, where the next request times are arranged in ascending order. We use $\mathbf{H}_i^{*}(l)$ and $\mathbf{H}_i^{A}(l)$ to represent the $l$-th element in these ordered sets.

**Lemma D.3.** *For an RB-compliant algorithm A, after serving each request $r_i$, $|\mathbf{H}_i^{*}| = |\mathbf{H}_i^{A}|$ and the following holds:*

$$\forall l = 1, ..., |\mathbf{H}_i^{*}| : \quad \mathbf{H}_i^{*}(l) \leq \mathbf{H}_i^{A}(l). \tag{6}$$

*Proof.* We prove this proposition by induction. Suppose the cache becomes full for the first time after serving $m$ page requests. The above inequalities hold trivially for any $i = 1, ..., m$, as no eviction occurs for either A or OPT. Now assume that after serving request $r_i$ ($i \geq m$), $|\mathbf{H}_i^{*}| \leq |\mathbf{H}_i^{A}|$ and (6) hold. We will show that they also hold after serving request $r_{i+1}$. There are three cases to consider when $r_{i+1}$ arrives:

**Case (i)** $p_{i+1} \in \mathbf{0}_i^{A}$. By Lemma D.2, $p_{i+1} \in \mathbf{0}_i^{*}$ also holds. Thus, $r_{i+1}$ results in a cache hit for both A and OPT. Then, $p_{i+1}$ remains in $\mathbf{0}_{i+1}^{A}$ and $\mathbf{0}_{i+1}^{*}$ if $y_{i+1} = 0$. Otherwise, $p_{i+1}$ is moved to $\mathbf{1}_{i+1}^{A}$ and $\mathbf{1}_{i+1}^{*}$ for A and OPT, respectively. In both scenarios, $|\mathbf{H}_{i+1}^{*}| \leq |\mathbf{H}_{i+1}^{A}|$ and element-wise relationship (6) is maintained after serving $r_{i+1}$.

**Case (ii)** $p_{i+1} \in \mathbf{0}_i^{*}$ but $p_{i+1} \notin \mathbf{0}_i^{A}$. Since $p_{i+1}$ is a 0-page at this moment, $p_{i+1} \notin \mathbf{1}_i^{A}$ also holds. By Lemma D.2, we know that $|\mathbf{0}_i^{A}| < |\mathbf{0}_i^{*}|$. Since $|\mathbf{0}_i^{A}| + |\mathbf{1}_i^{A}| = k = |\mathbf{0}_i^{*}| + |\mathbf{1}_i^{*}|$, we have $|\mathbf{1}_i^{*}| < |\mathbf{1}_i^{A}|$ and $|\mathbf{H}_i^{*}| < |\mathbf{H}_i^{A}|$. In this case, OPT experiences a cache hit while A suffers a cache miss.

Conceptually, cache evolvement for serving request $r_{i+1}$ includes two steps: (i) since A is RB-compliant, it must evict a page from $\mathbf{1}_i^{A}$; (ii) $p_{i+1}$'s next request time is either inserted to both $\mathbf{H}_{i+1}^{*}$ and $\mathbf{H}_{i+1}^{A}$ (if $y_{i+1} = 1$) or none (if $y_{i+1} = 0$). After these two steps, $|\mathbf{H}_{i+1}^{*}| \leq |\mathbf{H}_{i+1}^{A}|$ because $|\mathbf{H}_i^{*}| < |\mathbf{H}_i^{A}|$ before step (i), and step (ii) does not change the relation between $|\mathbf{H}_{i+1}^{*}|$ and $|\mathbf{H}_{i+1}^{A}|$. Moreover, the element-wise relationship (6) continues to hold for $i + 1$. This is because step (i) removes an element from $\mathbf{H}_i^{A}$, which can never decrease $\mathbf{H}_i^{A}(l)$ at any index $l$; and for step (ii), inserting the same element to both $\mathbf{H}_{i+1}^{*}$ and $\mathbf{H}_{i+1}^{A}$ does not change the element-wise relation.

**Case (iii)** $p_{i+1} \notin \mathbf{0}_i^*$. By Lemma D.2, we have $p_{i+1} \notin \mathbf{0}_i^A$. Note that $p_{i+1} \notin \mathbf{1}_i^*$ must hold because OPT evicts 1-pages at some time before they are requested again. Thus, OPT suffers a cache miss and evicts the 1-page with the furthest next request time from $\mathbf{1}_i^*$. Meanwhile, A can experience either a cache miss or a cache hit.

- If A experiences a cache hit, $p_{i+1} \in \mathbf{1}_i^A$ as $p_{i+1} \notin \mathbf{0}_i^A$. We can conceptually consider cache evolvement as two steps: (i) OPT evicts the 1-page with the furthest next request time from $\mathbf{1}_i^*$, and A evicts $p_{i+1}$ from $\mathbf{1}_i^A$; (ii) $p_{i+1}$ is either inserted to both $\mathbf{1}_{i+1}^*$ and $\mathbf{1}_{i+1}^A$ (if $y_{i+1} = 1$) or none (if $y_{i+1} = 0$). After these two steps, $|\mathbf{H}_{i+1}^*| \le |\mathbf{H}_{i+1}^A|$ apparently holds because $|\mathbf{H}_i^*| \le |\mathbf{H}_i^A|$ before step (i). Moreover, step (i) does not affect the element-wise relationship (6) because OPT evicts the page with the furthest next request time from $\mathbf{1}_i^*$, i.e., removing the last element from $\mathbf{H}_i^*$. For step (ii), inserting the same element to both $\mathbf{H}_{i+1}^*$ and $\mathbf{H}_{i+1}^A$ does not change the element-wise relation either. Therefore, (6) continues to hold for $i + 1$.

- If A suffers a cache miss, it just changes "A evicts $p_{i+1}$ from $\mathbf{1}_i^A$" in step (i) above to be "A evicts a page from $\mathbf{1}_i^A$" since A is RB-compliant. All the above arguments continue to apply, so both $|\mathbf{H}_{i+1}^*| \le |\mathbf{H}_{i+1}^A|$ and the element-wise relationship (6) for $i + 1$ hold.

By induction, $|\mathbf{H}_i^*| \le |\mathbf{H}_i^A|$ and (6) hold for all $i = 1, ..., n$, where $n$ denotes the total number of requests. This completes the proof. $\qquad\square$

**Proposition 3.5.** *For an RB-compliant algorithm* A*, after serving each request* $r_i$*, the next request time of any page in* $\mathbf{1}_i^A$ *is later than that of any page in* $\mathbf{0}_i^A$*.*

*Proof.* Suppose $p_x$ and $p_y$ are the 1-pages with the earliest next request times in $\mathbf{1}_i^A$ and $\mathbf{1}_i^*$, respectively. By Proposition D.3, the next request time of $p_x$ is not earlier than that of $p_y$. By Proposition D.1, the next request time of $p_y$ is later than any 0-page in $\mathbf{0}_i^*$. Thus, the next request time of $p_x$ is also later than any 0-page in $\mathbf{0}_i^*$. Additionally, we have $\mathbf{0}_i^A \subseteq \mathbf{0}_i^*$ by Lemma D.2. Therefore, the next request time of $p_x$ is later than any 0-page in $\mathbf{0}_i^A$, completing the proof. $\qquad\square$

# E    Proof of Lemma 3.8

**Lemma 3.8.** *RB-following algorithms that blindly follow predictions have unbounded robustness.*

*Proof.* Consider a cache of size 2, where at time $t$, pages $p_a$ and $p_b$ are cached. From time $t$ onward, an infinite sequence of alternating requests for $p_c$ and $p_b$ arrives. OPT incurs only one cache miss by evicting $p_a$ upon the first request for $p_c$, making $p_a$ a 1-page and both $p_b$ and $p_c$ 0-pages indefinitely. Suppose an RB-following algorithm A receives completely incorrect predictions. If A blindly trusts the predictions, it may instead evict $p_b$ and $p_c$ alternately, resulting in an infinite number of cache misses. Since OPT incurs only one miss, the competitive ratio of A is unbounded, implying infinite robustness. $\qquad\square$

# F    RB-Following Algorithms that Blindly Follow Predictions

Algorithm 2 gives a template of RB-following algorithms that blindly follow predictions. Refer to Sec. 3.2 for the definition of RB-following.

---
**Algorithm 2** An RB-following algorithm that blindly follows predictions
---
1: **for** $i = 1, ..., n$ **do**
2:     Receive the page request $r_i = (t_i, p_i)$ at time $t_i$
3:     **if** $p_i$ *is not in the cache* **then**
4:         **if** *the cache is full* **then**
5:             Call function `Evict()`
6:         **end if**
7:         Load $p_i$ into the cache
8:     **end if**
9: **end for**
---

Algorithm 3, 4, and 5 define the eviction functions of BLINDORACLE, LRB, and PARROT, respectively. They can be embedded into the template Algorithm 2 to restore the corresponding algorithms.

---
**Algorithm 3** `Evict()` of BLINDORACLE
---
1: Invoke the predictor to get the predicted next request times of cache pages
2: Evict the page with the furthest predicted next request time
---

---
**Algorithm 4** `Evict()` of LRB
---
1: Invoke the predictor to get the predicted Belady's binary labels of cached pages
2: **if** *there is a predicted 1-page* **then**
3:     Evict a predicted 1-page uniformly at random
4: **else**
5:     Evict a predicted 0-page uniformly at random
6: **end if**
---

---
**Algorithm 5** `Evict()` of PARROT
---
1: Invoke the predictor to get the predicted FitF page
2: Evict the predicted FitF page
---

## G   Proof of Eviction Pattern of RB-Compliant Algorithms

The following theorem corresponds to Observation 3 in Section 4.1.

**Theorem G.1.** *For an RB-compliant algorithm* A, *it never keeps an unrequested page in the cache during the time interval between the eviction and next request of another page. Formally, if a request $r_i$ results in a cache miss at time $t_i$, and the requested page $p_i$ was previously evicted at time $\mu < t_i$, then all pages remaining in the cache at time $t_i$ must have been requested at least once between $\mu$ and $t_i$.*

*Proof.* Suppose, for contradiction, that there is a cached page $p_\alpha$ at time $t_i$ that has not been requested since time $\mu$. Consider an alternative algorithm B, which behaves identically to A up to $t_i$, except that it evicts $p_\alpha$ instead of $p_i$ at time $\mu$, thereby keeping $p_\alpha$ cached until $t_i$. Algorithm B incurs the same number of cache misses as A before time $t_i$. However, at time $t_i$, B experiences a cache hit, whereas A incurs a cache miss due to the prior eviction of $p_i$. This indicates that B outperforms A after serving $r_i$, contradicting the optimality of A, as established in Proposition 3.4.   □

## H   Proof that LRB is RB-following

We first show a general conclusion below.

**Lemma H.1.** *If algorithm* A *is RB-following when serving $\{r_1, ..., r_j\}$, then* A&OPT$(j)$ *is RB-following. Here, we define the following: (1)* OPT(A, $j$) *denotes the* OPT *algorithm that begins with the cache content* A$(\{r_1, ..., r_j\})$ *and serves the subsequent requests in $\sigma$, where* A$(\{r_1, ..., r_j\})$

*represents the cache content of* A *after serving the first* $j$ *requests; (2)* A&OPT$(j)$ *denotes the hybrid algorithm that follows* A *when serving* $\{r_1, ..., r_j\}$ *and then follows* OPT$(A, j)$ *for the remaining sequence.*

*Proof.* Since A is RB-following before serving $r_{j+1}$, we have $\mathbf{0}_j^A = \mathbf{0}_j^*$ by Lemma 3.2 and Proposition 3.4, implying that OPT also incurs a cache miss at time $t_{j+1}$ and evicts a 1-page. Moreover, $|\mathbf{1}_j^A| = |\mathbf{1}_j^*| > 0$. By Belady's rule, at time $t_{j+1}$, OPT$(A, j)$ evicts the page with the furthest next request time, denoted as $p_\alpha$. By Proposition 3.5 and $|\mathbf{1}_j^A| > 0$, we have $p_\alpha \in \mathbf{1}_j^A$. This implies that A&OPT$(j)$ is RB-following until served $r_{j+1}$ as it prioritizes the eviction of 1-pages.

Therefore, if algorithm A is RB-following when serving $\{r_1, ..., r_j\}$, then A&OPT$(j)$ remains RB-following after serving $r_{j+1}$. By induction, A&OPT$(j)$ is RB-following for the entire request sequence, completing the proof. $\qquad\square$

**Proposition H.2.** LRB *is RB-following.*

*Proof.* Suppose LRB is RB-following when serving $\{r_1, ..., r_j\}$ and incurs a cache miss when serving request $r_{j+1}$. By Lemma H.1 and Corollary 3.3, LRB&OPT$(j)$ is RB-following and always evicts only 1-pages under perfect predictions. Recall that LRB relies on predicted 1-pages labeled by the OPT that begins with the current cache content of LRB, which corresponds exactly to LRB&OPT$(j)$. Consequently, under perfect predictions, all predicted 1-pages are 1-pages labeled by OPT. Therefore, LRB remains RB-following after serving $r_{j+1}$. The proof follows by induction. $\quad\square$

# I   Proof of Lemma 4.3

**Lemma 4.3.** $\sum_{q \in \mathcal{Q}} \frac{1}{2} c_q \leq \text{OPT} \leq \sum_{q \in \mathcal{Q}} n_q^{old}$.

*Proof.* We first show $\text{OPT} \leq \sum_{q \in \mathcal{Q}} n_q^{old}$. Assume an alternative algorithm GUARD&B that behaves identically to GUARD&A except that, when serving requests for new pages, it evicts the $n_q^{old}$ old pages whose next request times are later than those of the remaining old pages. As a result, GUARD&B experiences cache hits when the remaining old pages are requested. Note that at the end of the $q$-th phase, the cache content of GUARD&A is identical to that of GUARD&B. Clearly, the total cost of GUARD&B is $\sum_{q \in \mathcal{Q}} n_q^{old}$, implying that $\text{OPT} \leq \sum_{q \in \mathcal{Q}} n_q^{old}$.

Next, the proof of $\text{OPT} \geq \sum_{q \in \mathcal{Q}} \frac{1}{2} c_q$ follows from Fiat et al. [3], though our phase definition differs from theirs.

Let $\text{OPT}_q$ represent the number of cache misses incurred by OPT while processing the requests that arrive during the $q$-th phase of GUARD&A. Let $\mathcal{S}_q^*$ and $\mathcal{S}_q^{\text{GA}}$ denote the sets of cached pages for OPT and GUARD&A, respectively, at the beginning of the $q$-th phase. Define $h_q$ $(0 \leq q \leq Q)$ as the number of pages in OPT's cache but absent in GUARD&A's cache, i.e., $h_q = k - |\mathcal{S}_q^* \cap \mathcal{S}_q^{\text{GA}}|$. Note that $h_0 = 0$.

First, we claim that $\text{OPT}_q \geq c_q - h_q$. Among the $c_q$ distinct new page requests in the $q$-th phase, at most $h_q$ of them may already reside in OPT's cache. Hence, OPT incurs at least $c_q - h_q$ cache misses.

Next, we show that $\text{OPT}_q \geq h_{q+1}$. By definition, the pages in GUARD&A's cache at the end of the $q$-th phase, $\mathcal{S}_{q+1}^{\text{GA}}$, remain in the cache at the beginning of the $(q+1)$-th phase. Since OPT's cache $\mathcal{S}_{q+1}^*$ contains $h_{q+1}$ pages not present in $\mathcal{S}_{q+1}^{\text{GA}}$, it follows that $\mathcal{S}_{q+1}^{\text{GA}}$ also contains $h_{q+1}$ pages that are absent in $\mathcal{S}_{q+1}^*$. Moreover, since every cached page in $\mathcal{S}_{q+1}^{\text{GA}}$ is requested at least once during the $q$-th phase, OPT must have evicted at least $h_{q+1}$ pages within the same phase.

Note that the final phase ($Q$-th phase) may not end after serving all requests. Thus, to be rigorous, we denote $h_{Q+1}$ as the number of pages that satisfy two conditions: (1) The page has been requested at least once in the $Q$-th phase. (2) The page is in GUARD&A's cache but absent in OPT's cache after

serving all requests. Similarly, OPT must have evicted at least $h_{Q+1}$ ($h_{Q+1} > 0$) pages within the same phase.

Combining the two inequalities, we have:

$$\sum_{q=0}^{Q} \text{OPT}_q \geq \sum_{q=0}^{Q} \max\left(c_q - h_q, h_{q+1}\right)$$

$$\geq \sum_{q=0}^{Q} \frac{1}{2}\left(c_q - h_q + h_{q+1}\right)$$

$$= \sum_{q=0}^{Q} \frac{1}{2}c_q - h_0 + h_{Q+1}$$

$$\geq \sum_{q=0}^{Q} \frac{1}{2}c_q \quad (\text{since } h_0 = 0, h_{Q+1} \geq 0), \tag{7}$$

which completes the proof. $\square$

## J    A Tight Analysis of GUARD&A's Robustness

**Theorem 4.6.** GUARD&A *is* $(2H_{k-1} + 2)$-*robust, which is tight.*

*Proof.* We adopt Lemma 4.3, Lemma 4.4, and notations from Sec. 4.3.

All evictions that occur during a given phase form an eviction graph. A directed edge $(p_\alpha, p_\beta)$ is added from page $p_\alpha$ to page $p_\beta$ if $p_\alpha$ is evicted upon the arrival of request $r_\beta$, which targets page $p_\beta$. At a high level, the directed edge represents "evicted by". Figure 3 illustrates an example of such a graph consisting of multiple connected subgraphs. In the graph, edges with "pred." and "rand." denote prediction-driven evictions and random evictions, respectively.

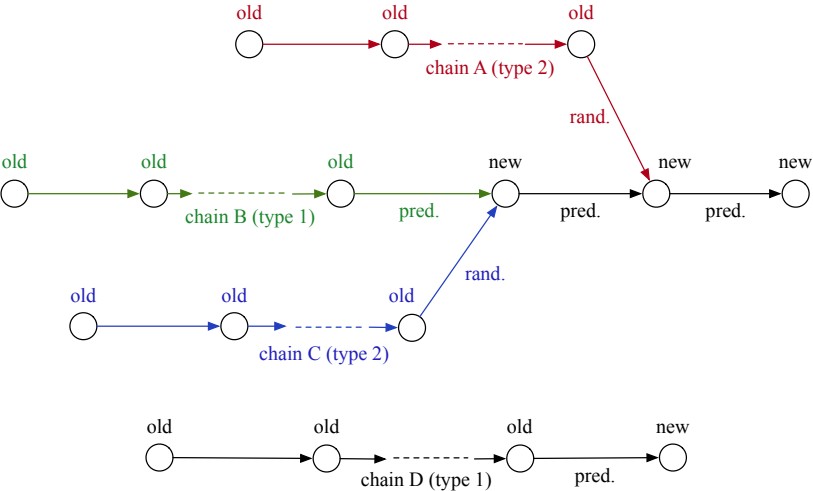

Figure 3: Eviction Graph

The number of edges in the eviction graph equals the number of evictions that occur during the phase. According to the definition of GUARD&A, an eviction graph has the following properties:

1. The out-degree of any page is at most 1, as each page can be evicted no more than once within the phase under the "guard" mechanism.

2. The in-degree of each old page is at most 1, as it will be guarded after being loaded into the cache.

3. The in-degree of each new page is at most 2, as it can be loaded into the cache at most twice. In addition, a new page with an in-degree of 2 must have an out-degree of 1. It evicts a page based on prediction upon its first cache miss, and a randomly selected old page upon its second cache miss.

4. A new page can only point to (be evicted by) another new page (based on predictions), since a request for an old page or a previously evicted new page only leads to the random eviction of an old page upon a cache miss.

5. A page with an out-degree of 0 must be a new page, as an old page remains in the cache if it is not evicted during the phase.

A chain that contains multiple old pages and a single new page is defined as an old-page eviction chain, such as chains A, B, C, and D in Figure 3. The following two conclusions hold regarding the length of an old-page eviction chain, which is defined as the number of edges and depends on the chain type. We classify old-page eviction chains based on the first eviction.

- **Type 1:** the request for the new page evicts an old page based on predictions. The expected length of this type of chain is at most $H_{k-1} + 1$.

  *Proof.* Denote $L(x)$ as the expected number of subsequent random evictions of old pages when the number of unrequested old pages in the cache is $x$. This implies that, subsequently, up to $x$ old pages will be requested during this phase. Thus, we have

  $$L(x) \leq 1 + \frac{1}{x} \sum_{i=0}^{x-1} L(i), \tag{8}$$

  where $L(0) = 0$. Consequently, $L(x) \leq \sum_{i=1}^{x} \frac{1}{i} = H_x$.

  In the worst case, the old page with the earliest next request time is evicted by the request for the new page. Old pages are evicted randomly afterwards, so the expected length of the sub-chain is bounded by $L(k-1) + 1 = H_{k-1} + 1$. $\qquad\square$

- **Type 2:** the request for the new page evicts an old page uniformly at random. The expected length of this type of chain is at most $H_{k-1}$.

  *Proof.* When the request for the new page evicts an old page randomly, at most $k - 1$ unrequested old pages remain in the cache, since the new page must have been evicted previously. Thus, the expected length of the chain is at most $L(k-1) = H_{k-1}$. $\qquad\square$

Within a connected subgraph with $x$ distinct new pages, all new pages form a chain, according to Properties 1 and 4 stated above. Based on Property 3, at most one old-page-eviction chain of type 1 and $x - 1$ old-page-eviction chains of type 2 exist in the subgraph. See Figure 3 for an illustration. Therefore, the expected number of edges within this subgraph is at most $H_{k-1} + 1 + (x-1)H_{k-1} + (x-1) = x(H_{k-1} + 1)$.

By Lemma 4.4, we know that the number of old-page eviction chains $n_q^{old}$ is at most $c_q$. This leads to the following lemma directly.

**Lemma J.1.** *At most $c_q$ old-page eviction chains exist in the eviction graph of phase $q$, where $c_q$ denotes the number of distinct new pages requested during this phase.*

Combining the above conclusions, the expected number of evictions in phase $q$ is at most $c_q(H_{k-1}+1)$. Finally, we derive an upper bound on the total cost of GUARD&A as follows.

$$\mathbb{E}[\text{GA}] = d_0 + \sum_{q=1}^{Q} \mathbb{E}[\text{GA}_q] \leq (H_{k-1} + 1) \sum_{q=0}^{Q} c_q$$

$$\leq (2H_{k-1} + 2)\text{OPT}, \quad \text{(by Lemma 4.3)} \tag{9}$$

where both GUARD&A and OPT experience $d_0$ cache misses during the 0-th phase as no evictions occur. GUARD&A is $(2H_{k-1} + 2)$-robust, as established by (4.3).

**Tightness.** Consider the case where $n_q$ evictions occur at the beginning of the phase, evicting exactly $c_q$ old pages (i.e., $n_q = c_q$) whose next request times are earlier than the remaining old pages. In this case, we can bound GUARD&A's total cost GA as follows, where $j$ denotes the $j$-th subsequent distinct old page request.

$$
\begin{aligned}
\mathbb{E}[\text{GA}] = c_0 + \sum_{q=1}^{Q} \mathbb{E}[\text{GA}_q] &= c_0 + \sum_{q=1}^{Q} \Big( n_q + \mathbb{E}[o_q] \Big) \\
&= c_0 + \sum_{q=1}^{Q} \Big( n_q + c_q + \sum_{j=c_q+1}^{k-c_q} \frac{c_q}{k - (j-1)} \Big) \\
&= c_0 + \sum_{q=1}^{Q} (2 + H_{k-c_q} - H_{c_q}) c_q \quad \text{(since } n_q = c_q) \\
&\leq c_0 + \sum_{q=1}^{Q} (H_{k-1} + 1) c_q. \quad \text{(equality holds if } c_q = 1)
\end{aligned}
$$

(10)

By (10), we have

$$
\begin{aligned}
\lim_{Q \to \infty} \mathbb{E}[\text{GA}] &\leq \sum_{q \in \mathcal{Q}} (H_{k-1} + 1) c_q \\
&\leq (2H_{k-1} + 2) \text{OPT}. \quad \text{(by Lemma 4.3)}
\end{aligned}
$$

(11)

We now demonstrate that the equality in the final inequality (11) can indeed be achieved under the following conditions, where all previously defined settings remain unchanged.

Let $w_q$ denote the only new page requested during phase $q$, implying that $c_q = 1$. Consider the cache contents of GUARD&A and OPT after all requests in phase $q$ have been served:

- GUARD&A: GUARD&A's cache contains $w_q$ and $k - 1$ old pages at the end of phase $q$. This is because the request for $w_q$ evicts the old page with the earliest next request time, leading to subsequent random evictions of only old pages.

- OPT: Assume that OPT's cache content at the beginning of phase $q$ matches that of GUARD&A and that $w_q$ is never requested again after its next request. Following Belady's rule, OPT evicts $w_q$ during this phase. All subsequent old page requests result in cache hits. Consequently, at the end of phase $q$, OPT's cache contains all old pages, with one old page not being requested during phase $q$. We denote this old page as $u_q$.

As a result, at the start of phase $q + 1$, $u_q$ resides in OPT's cache but is absent from GUARD&A's cache. Suppose $u_q$ serves as the only new page in phase $q + 1$, implying $c_{q+1} = 1$, and is requested before any other page in phase $q + 1$. After serving requests to $k - 1$ distinct old pages, phase $q + 1$ ends. At this point, both OPT and GUARD&A once again have identical cache contents.

This construction enables repeated alternation between the behaviors in phases $q$ and $q + 1$ for all subsequent phases. Notably, OPT incurs a total cost of 1 across phases $q$ and $q + 1$, which equals $(c_q + c_{q+1})/2$. Consequently, as this pair of phases repeats and $Q \to \infty$, we obtain:

$$
\text{OPT} = \sum_{q \in \mathcal{Q}} \frac{1}{2} c_q,
$$

thereby establishing the tightness of the bound in 5. This case confirms the tightness.

$\square$

# K    Applications of GUARD

## K.1    Algorithm

We first present the pseudo-codes for the three variants of GUARD&A.

---
**Algorithm 6** GUARD&A
---
1: $\mathcal{U} \leftarrow \emptyset$ (the set tracks *unrequested old pages* in the cache)
2: **for** $i = 1, ..., n$ **do**
3:     Receive a page request $r_i = (t_i, p_i)$ at time $t_i$
4:     **if** $p_i$ *is not in the cache* **then**
5:         **if** *the cache is full* **then**
6:             **if** $\mathcal{U}$ is empty **then**
7:                 *Unguard all cached pages*
8:                 $\mathcal{U} \leftarrow$ {all cached pages} *(a new phase begins)*
9:             **end if**
10:            **if** $p_i$ *was evicted in the current phase* **then**
11:                Evict a page $x$ from $\mathcal{U}$ uniformly at random
12:                *Guard* $p_i$
13:            **else**
14:                $\mathcal{S} \leftarrow$ {all unguarded cached pages}
15:                Call function $\texttt{Evict}(\mathcal{S})$
16:            **end if**
17:            **if** the evicted page $x \in \mathcal{U}$ **then**
18:                $\mathcal{U} \leftarrow \mathcal{U} \backslash \{x\}$
19:            **end if**
20:        **end if**
21:        Load $p_i$ into the cache
22:    **end if**
23:    **if** $p_i \in \mathcal{U}$ **then**
24:        $\mathcal{U} \leftarrow \mathcal{U} \backslash \{p_i\}$
25:    **end if**
26: **end for**
---

Algorithms 7, 8, and 9 define the eviction functions of GUARD&BLINDORACLE, GUARD&LRB, and GUARD&PARROT, respectively. They can be embedded into Algorithm 6 to restore the corresponding algorithms.

---
**Algorithm 7** $\texttt{Evict}(\mathcal{S})$ of GUARD&BLINDORACLE
---
1: Invoke the predictor to get the predicted next request times of pages in $\mathcal{S}$
2: Evict the page with the furthest predicted next request time from $\mathcal{S}$
---

---
**Algorithm 8** $\texttt{Evict}(\mathcal{S})$ of GUARD&LRB
---
1: Invoke the predictor to get the predicted Belady's binary labels of pages in $\mathcal{S}$
2: **if** *there is a predicted 1-page* **then**
3:     Evict a predicted 1-page from $\mathcal{S}$ uniformly at random
4: **else**
5:     Evict a predicted 0-page from $\mathcal{S}$ uniformly at random
6: **end if**
---

---
**Algorithm 9** $\texttt{Evict}(\mathcal{S})$ of GUARD&PARROT
---
1: Invoke the predictor to get the predicted FitF page from $\mathcal{S}$
2: Evict the predicted FitF page from $\mathcal{S}$
---

All these algorithms invoke the predictor upon eviction, consistent with common implementations in real systems. GUARD&A restricts the range of predictions only for unguarded pages in the cache, denoted by $\mathcal{S}$. Below, we present implementation details regarding the predictor queries.

- GUARD&BLINDORACLE requires only the predicted next request times for pages in $\mathcal{S}$, which is straightforward to implement since each prediction is independent.

- GUARD&LRB utilizes the predicted Belady's binary labels only for pages in $\mathcal{S}$, as labeled by the OPT algorithm that begins with the current cache content and serving subsequent requests in $\sigma$ without evicting pages outside of $\mathcal{S}$. This is achieved by the LRB Predictor [9], which first predicts the next request times of pages in $\mathcal{S}$ and then classifies them based on the predicted Belady boundary.

- GUARD&PARROT queries the predictor for the FitF page within $\mathcal{S}$, selected based on the eviction priorities assigned by the FitF page predictor, a neural network-based model proposed by Liu et al. [10].

### K.2  Smoothness Proof for GUARD&BLINDORACLE

We first show the relationship between prediction error $\eta_t$ and inversions, as studied by Rohatgi [6].

**Definition K.1.** *(Inversion). Given two sequences $A = (a_1, a_2, ..., a_n)$ and $B = (b_1, ..., b_n)$, let $inv(A, B)$ be the number of inversions, which denotes the number of pairs of indices $(i, j)$ such that $a_i < a_j$ but $b_i > b_j$.*

**Definition K.2.** *($l_1$-distance). Given two sequences $A = (a_1, ..., a_n)$ and $B = (b_1, ..., b_n)$, we define $l_1(A, B) = \sum_{i=1}^{n} |a_i - b_i|$ as the $l_1$-distance between $A$ and $B$.*

We then present a lemma from Rohatgi [6], with its descriptive form slightly modified for clarity.

**Lemma K.3.** *(Rohatgi [6], Lemma 4.1) Let $A = (a_1, ..., a_n)$ and $B = (b_1, ..., b_n)$ be two integer sequences. Then $inv(A, B) \leq 2l_1(A, B)$, with $inv(A)$ and $l_1$ as defined above.*

Next, we relate $\eta_t$ to the number of inversions. For a page request sequence $\sigma$, let their next request time be $\theta = \{T_1, T_2, ..., T_n\}$, and let $\hat{\theta} = \{\hat{T}_1, ..., \hat{T}_n\}$ represent the predicted arrival times. By definition, $\eta_t = l_1(\theta, \hat{\theta})$. Then, by Lemma K.3, we have

$$inv(\theta, \hat{\theta}) \leq 2l_1(\theta, \hat{\theta}) = 2\eta_t \tag{12}$$

**Theorem K.4.** GUARD&BLINDORACLE *is $\mathcal{O}(\log(\eta_t/\text{OPT}))$-smooth.*

*Proof.* In the following, we bound the expected length of an old-page eviction chain in phase $q$, called $\mathcal{E}_q$, according to the properties of the eviction graph (Figure 3). Denote $|\mathcal{E}_q|$ as its length.

- If $\mathcal{E}_q$ is a chain of type 1, which consists of a new page and multiple old pages, and the request for the new page evicts an old page based on predictions. Denote the new page as $p_\alpha$. Let $\mathcal{U}_\beta$ be the set of unrequested old pages when the old page $p_\beta$, which was previously evicted at time $\mu$ by the request for $p_\alpha$, is requested and a cache miss occurs. This contributes at least $I(\mathcal{E}_q) = |\mathcal{U}_\beta|$ inversions, since each pair of $p_\beta$ and a page in $\mathcal{U}_\beta$ indicates an inversion in predictions at time $\mu$.

  According to $L(x) \leq H_x$ in Appendix J, we have $\mathbb{E}[|\mathcal{E}_q|] \leq \mathcal{O}(\log I(\mathcal{E}_q))$.

- If $\mathcal{E}_q$ is a chain of type 2, which consists of a new page and multiple old pages, and the request for the new page $p_\alpha$ randomly evicts an old page. Its new page $p_\alpha$ must be previously evicted by another new page $p_\gamma$, as $p_\alpha$ must have an in-degree of 2 and an out-degree of 1 in the eviction graph, according to Property 3 mentioned in Appendix J. Let $\mathcal{U}_\alpha$ be the set of unrequested old pages when the new page $p_\beta$ is requested. Each pair of $p_\alpha$ and a page in $\mathcal{U}_\alpha$ indicates an inversion in predictions when evicting $p_\alpha$. Similarly, this chain contributes at least $I(\mathcal{E}_q) = |\mathcal{U}_\alpha|$ inversions, and we have $\mathbb{E}[|\mathcal{E}_q|] \leq \mathcal{O}(\log I(\mathcal{E}_q))$.

The total inversions $\text{inv}(\theta, \hat{\theta}) \geq \sum_{q \in \mathcal{Q}} \sum_{\mathcal{E}_q} I(\mathcal{E}_q)$. Suppose there are a total of $N$ old-page eviction chains across all phases, where $N = \sum_{q \in \mathcal{Q}} \sum_{\mathcal{E}_q} 1 = \sum_{q \in \mathcal{Q}} n_q^{old}$ by definition. By Lemma 4.3, we have

$$\text{OPT} \leq N = \sum_{q \in \mathcal{Q}} n_q^{old} \leq \sum_{q \in \mathcal{Q}} c_q \leq 2\text{OPT}. \tag{13}$$

GUARD&BLINDORACLE's expected total cost $\mathbb{E}[\text{G\&B}]$ is:

$$\begin{aligned}
\mathbb{E}[\text{G\&B}] &\leq \sum_{q \in \mathcal{Q}} \Big( c_q + \sum_{\mathcal{E}_q} \mathbb{E}[|\mathcal{E}_q|] \Big) \\
&\leq \sum_{q \in \mathcal{Q}} c_q + \sum_{q \in \mathcal{Q}} \sum_{\mathcal{E}_q} \mathcal{O}(\log I(\mathcal{E}_q)) \quad \text{(by Lemma 4.3)} \\
&\leq \sum_{q \in \mathcal{Q}} c_q + N \cdot \mathcal{O}(\log(\frac{\text{inv}(\theta, \hat{\theta})}{N})) \quad \text{(by Jensen's inequality and concavity)} \\
&\leq 2\text{OPT} + 2\text{OPT} \cdot \mathcal{O}(\log(\frac{\eta_t}{\text{OPT}})) \quad \text{(by (13) and Lemma K.3)} \\
&= \mathcal{O}(\text{OPT} \cdot \log(\frac{\eta_t}{\text{OPT}})). \tag{14}
\end{aligned}$$

This demonstrates the $\mathcal{O}(\log(\eta_t/\text{OPT}))$-smoothness of GUARD&BLINDORACLE. □

### K.3 Smoothness Proof for GUARD&LRB

We first present a key lemma that helps establish smoothness. We denote OPT $(A, j)$ as the OPT algorithm that begins with the cache content $A(\{r_1, ..., r_j\})$ and serves the subsequent requests in $\sigma$ without evicting guarded pages. Here, $A(\{r_1, ..., r_j\})$ represents the cache content of A after serving the first $j$ requests.

**Lemma K.5.** *When serving request $r_{j+1}$ at time $t_{i+1}$, if GUARD&LRB evicts a 1-page $p_x$ (as labeled by OPT(A, j)), then the current phase must terminate before $p_x$ is requested again.*

*Proof.* Suppose OPT(A, $j$) evicts page $p_x$ at time $\mu > t_{i+1}$. By definition, $p_x$ is the page with the furthest next request time at time $\mu$. Let $T_x$ denote the next request time of $p_x$. This implies that the remaining $k-1$ pages in the cache at time $\mu$ will all be requested before $T_x$.

Whenever one of these remaining pages, say $p_\alpha$, is requested after time $\mu$, if it results in a cache hit, then $p_\alpha \notin \mathcal{U}$ or $\mathcal{U} = \mathcal{U} \setminus \{p_\alpha\}$. Otherwise, if a cache miss occurs, GUARD&LRB evicts a random page from $\mathcal{U}$. Therefore, after evicting $p_x$ and serving requests for all of these $k-1$ pages, the current phase must terminate. □

**Theorem K.6.** GUARD&LRB *is $\mathcal{O}(H_k \cdot \eta_b/\text{OPT})$-smooth.*

*Proof.* According to the description of GUARD&LRB provided in Section K.1, there is at least one 1-page in the set of unguarded pages $\mathcal{S}$ upon a cache miss. Otherwise, the OPT algorithm, starting from the current cache content, would have to evict a 0-page from $\mathcal{S}$, contradicting the definition of Belady's binary labels. Thus, the algorithm always evicts a 1-page when predictions are accurate upon a cache miss.

According to the properties of the eviction graph (Figure 3), each old-page eviction chain results from a unique prediction-driven eviction. By Lemma K.5, evicting a 1-page leads to no subsequent evictions. Thus, given a prediction error of $\eta_b$, the total expected length of all old-page eviction chains is at most $\mathcal{O}(H_k \cdot \eta_b)$, where each chain has length at most $\mathcal{O}(H_k)$ as shown in Appendix J.

$\mathcal{O}(H_k \cdot \eta_b)$ also bounds the expected total cost of GUARD&LRB, thereby completing the proof. □

### K.4 Smoothness Proof for GUARD&PARROT

**Theorem K.7.** GUARD&PARROT *is $\mathcal{O}(H_k \cdot \eta_f/\text{OPT})$-smooth.*

*Proof.* According to the properties of the eviction graph (Figure 3), each incorrect prediction of a FitF page leads to either (1) the eviction of an old page, which subsequently generates an old-page eviction chain of type 1, or (2) the eviction of a new page, which subsequently generates an old-page eviction chain of type 2.

Therefore, given a prediction error of $\eta_f$, there are at most $\eta_f$ old-page eviction chains in the eviction graphs across all phases. From Appendix J, the expected length of an old-page eviction chain is at most $\mathcal{O}(H_k)$. Thus, we can bound the expected total cost of GUARD&PARROT, denoted $\mathbb{E}[G\&P]$, as:

$$\mathbb{E}[G\&P] \leq \mathcal{O}(H_k \cdot \eta_f), \tag{15}$$

which completes the proof. □

## L  Lower Bounds

**Robustness.** No randomized learning-augmented can achieve robustness better than $H_k$, which is the lower bound of randomized online caching algorithms [3]. To the best of our knowledge, existing online algorithms that achieve $H_k$ competitive ratio include EQUITABLE [4], PARTITION [44], ONLINEMIN [45], among which ONLINEMIN has the lowest asymptotic time complexity and uses $\mathcal{O}(\log k)$ time per request (or $\mathcal{O}(\log k / \log \log k)$ in the RAM model). Thus, we may be able to further enhance the consistency/robustness trade-off of the robustification framework to $(1, H_k + \mathcal{O}(1))$, at the expense of more than $\mathcal{O}(1)$ additional computation per request.

**Smoothness.** Rohatgi [6] demonstrates that for a randomized learning-augmented caching algorithm utilizing next request time (NRT) predictions, the smoothness cannot be better than $\mathcal{O}(\log(1/(k \log k) \cdot \eta_t / \text{OPT}))$. To the best of our knowledge, no existing algorithm has achieved this bound. At the same time, whether improving smoothness to approach this bound compromises 1-consistency or bounded robustness remains an open problem.

## M  More Experimental Results and Discussions

### M.1  Experiments on Brightkite and Citi Using PLECO and POPU

We evaluate algorithms that utilize page request time predictions on the BrightKite and Citi datasets using PLECO and POPU, two widely adopted predictors in this domain [5, 20, 8]. For F&R, predictions of page request times are converted into action predictions, following the methodology of Sadek and Elias [8]. Table 5 presents the cost ratios of different algorithms. The results indicate that GUARD&B.O. consistently achieves the best or near-best performance.

Table 5: Cost ratios of algorithms on BrightKite and Citi using PLECO and POPU predictors.

| Dataset | Predictor | B.O. | P.M. | LM. | LNONM. | B.O.&M.$^D$ | F&R | GUARD&B.O. |
|---------|-----------|------|------|------|--------|-------------|------|------------|
| BrightKite | PLECO | 2.081 | 1.341 | 1.337 | 1.321 | 1.317 | 1.348 | **1.303** |
|  | POPU | 1.707 | 1.262 | 1.264 | 1.259 | 1.305 | 1.304 | **1.198** |
| Citi | PLECO | 2.277 | 1.877 | 1.875 | 1.888 | **1.860** | 1.864 | 1.900 |
|  | POPU | 1.739 | 1.776 | 1.779 | 1.769 | 1.732 | 1.790 | **1.693** |

### M.2  Experiments on SPEC CPU2006 Benchmark Using PLECO and POPU

To evaluate the performance of algorithms that utilize predictions of page request times in a more realistic scenario, we employ real-world memory trace datasets from the SPEC CPU2006 benchmark, following the setup of Chłędowski et al. [15].

**Average Cost Ratios.** Table 6 presents the average cost ratios of various algorithms relative to OPT across all 13 datasets. The results indicate that GUARD&B.O. achieves the best overall performance on average.

Table 6: Average cost ratios on SPEC CPU2006 Benchmark using PLECO and POPU predictors.

| Predictor | LRU | B.O. | P.M. | LM. | LNonM. | B.O.&M.$^D$ | F&R | Guard&B.O. |
|-----------|-----|------|------|-----|--------|-------------|-----|------------|
| **PLECO** | 1.478 | 1.404 | 1.335 | 1.335 | 1.346 | 1.294 | 1.360 | **1.226** |
| **POPU** | 1.478 | 1.261 | 1.312 | 1.320 | 1.312 | 1.233 | 1.319 | **1.203** |

**LRU-normalized Cost Ratios and Figures.** To ensure clear comparisons across different datasets, we adopt the LRU-normalized empirical competitive ratio from Chłędowski et al. [15], referred to as the LRU-normalized cost ratio (LCR) in this study. LCR evaluates an algorithm ALG in relation to LRU, and is defined as follows:

$$\text{LCR(ALG)} = \frac{\text{ALG} - \text{OPT}}{\text{LRU} - \text{OPT}}.$$

The average LRU-normalized cost ratios across all 13 datasets are summarized in Table 7, while Figures 4 and 5 provide a dataset-wise performance breakdown. These results further validate that Guard&B.O. consistently outperforms other methods or remains among the best-performing algorithms.

Table 7: Average LRU-normalized cost ratios on SPEC CPU2006 Benchmark using PLECO and POPU predictors.

| Predictor | LRU | B.O. | P.M. | LM. | LNonM. | B.O.&M.$^D$ | F&R | Guard&B.O. |
|-----------|-----|------|------|-----|--------|-------------|-----|------------|
| **PLECO** | 1 | 1.050 | 0.947 | 0.943 | 0.926 | 0.763 | 0.845 | **0.625** |
| **POPU** | 1 | 0.768 | 0.881 | 0.885 | 0.880 | 0.664 | 0.829 | **0.568** |

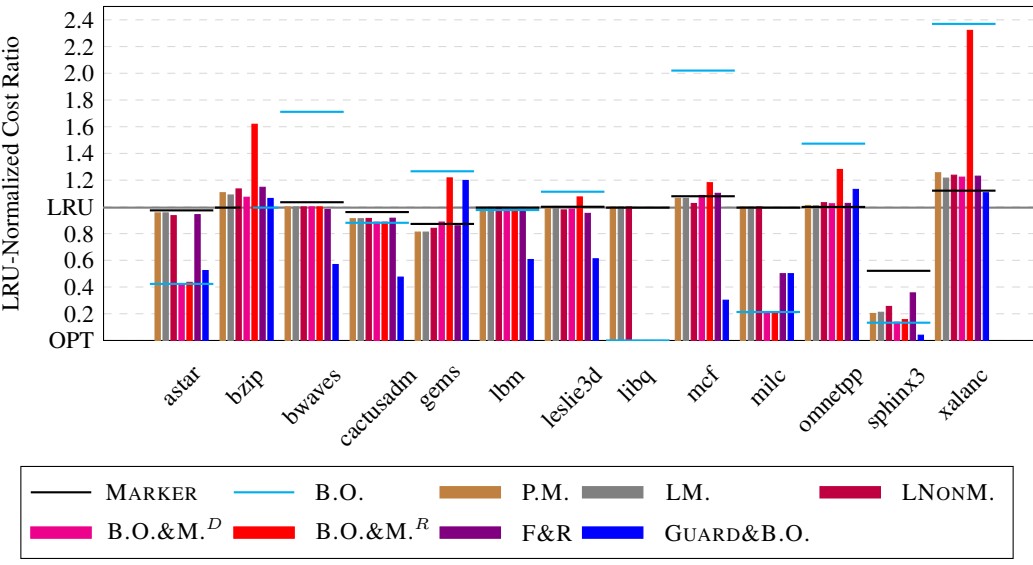

Figure 4: LRU-normalized cost ratios on SPEC CPU2006 Benchmark using the PLECO predictor.

## M.3 Experiments on SPEC CPU2006 Benchmark Using LRB predictor

In this section, we evaluate the performance of algorithms leveraging binary predictions of Belady's binary labels on datasets from the SPEC CPU2006 benchmark, using LRB predictor. The algorithm Mark&Predict (M.&P.) is excluded from this comparison, as it employs a different interpretation of predicted binary labels. In addition, we implement LRB&Marker$^D$ (LRB&M.$^D$)

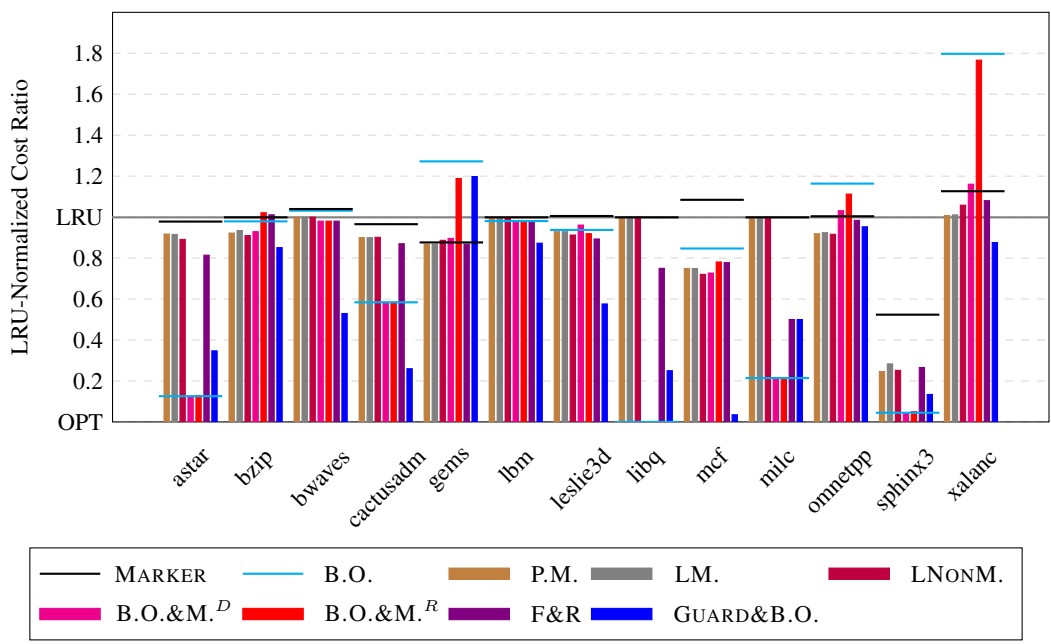

Figure 5: LRU-normalized cost ratios on SPEC CPU2006 Benchmark using the POPU predictor.

and LRB&MARKER$^R$ (LRB&M.$^R$), which combine LRB with MARKER using deterministic and randomized switching, respectively, following the switching-based approach introduced by Wei [7].

Following the methodology of Song et al. [9], we extract features from the SPEC CPU2006 datasets, setting $|\mathtt{Delta}_i| = 10$ and $|\mathtt{EDC}_i| = 10$. In addition to the original PC and address features, a total of 22 features were used for training. The GBM model is configured with a learning rate of 0.01, a maximum depth of 6, and 31 leaves. Both the sub-sample rate and column sample rate are set to 0.8. The model employs L2-norm loss, and early stopping is applied at 8000 rounds to determine the optimal parameters.

**Average Cost Ratios.** Table 8 presents the average cost ratios of all evaluated algorithms relative to OPT across all 13 datasets. The results show that GUARD&LRB outperforms other algorithms, including the base algorithm LRB.

Table 8: Average cost ratios on SPEC CPU2006 Benchmark using LRB predictor.

| Predictor | LRU | MARKER | LRB | MARK0 | LRB&M.$^D$ | LRB&M.$^R$ | GUARD&LRB |
|---|---|---|---|---|---|---|---|
| LRB predictor | 1.478 | 1.394 | 1.281 | 1.268 | 1.259 | 1.293 | **1.171** |

**LRU-normalized Cost Ratios and Figure.** Table 9 and Figure 6 show the LCR comparison when using GBM.

Table 9: Average LRU-normalized cost ratios on SPEC CPU2006 Benchmark using LRB predictor.

| Predictor | LRU | MARKER | LRB | MARK0 | LRB&M.$^D$ | LRB&M.$^R$ | GUARD&LRB |
|---|---|---|---|---|---|---|---|
| LRB predictor | 1 | 0.970 | 0.809 | 0.506 | 0.714 | 0.808 | **0.357** |

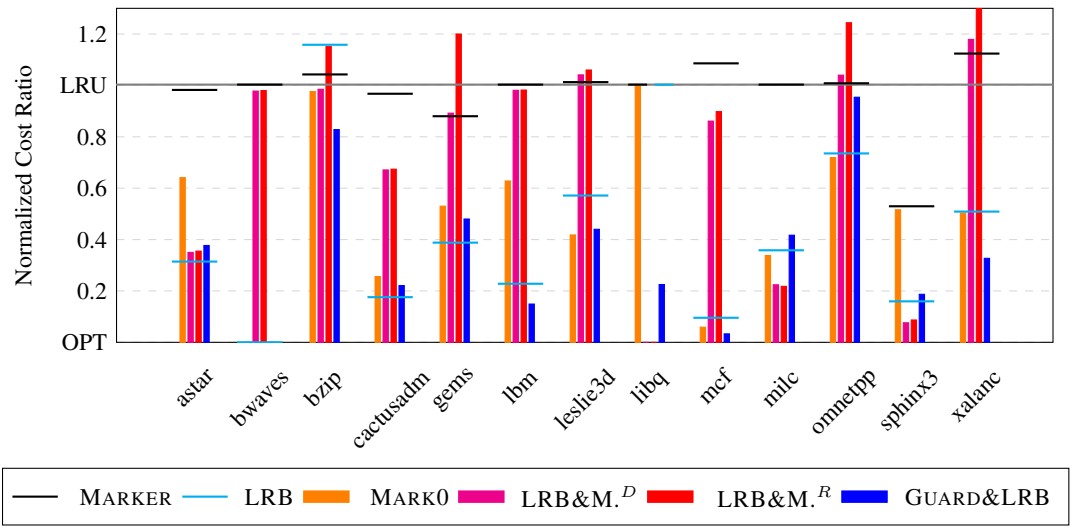

Figure 6: LRU-normalized cost ratios on SPEC CPU2006 Benchmark using LRB predictor.

## M.4 Experiments on SPEC CPU2006 Benchmark Using Parrot

**FitF page predictor.** We investigate Parrot, a neural network-based model proposed by Liu et al. [10], which utilizes an imitation learning approach to approximate the optimal policy. Built on LSTM and attention mechanisms, Parrot predicts the page with the highest eviction priority whenever an eviction is needed. In our experiments, Parrot is trained for 20,000 steps with a batch size of 32, without applying the Dagger algorithm [46].

**The Algorithm PARROT.** We propose PARROT (abbreviated as PA.), which directly follows the prediction of the Parrot model and evicts the predicted FitF page. PARROT is also an RB-following algorithm that blindly follows predictions. It is clear that PARROT is 1-consistent, while GUARD&PARROT (abbreviated as GUARD&PA.) is $(2H_{k-1} + 2)$-robust. Additionally, we implement PARROT&MARKER$^D$ and PARROT&MARKER$^R$, which combine PARROT with MARKER using deterministic and randomized switching, respectively, following the switching-based approach introduced by Wei [7].

Interestingly, the best algorithms for the two metrics differ due to their different calculations. Nonetheless, GUARD&PARROT consistently achieves either the best or near-best performance.

**Average Cost Ratios.** Table 10 shows the average cost ratios of algorithms compared to OPT across all 13 datasets.

Table 10: Average cost ratios on SPEC CPU2006 Benchmark using the FitF page predictor.

| Predictor | LRU | MARKER | PA. | PA.&M.$^D$ | PA.&M.$^R$ | GUARD&PA. |
|---|---|---|---|---|---|---|
| FitF page predictor | 1.478 | 1.394 | 1.363 | **1.315** | 1.348 | 1.338 |

**LRU-normalized Cost Ratios and Figure.** Table 11 and Figure 7 show the LCR comparison.

Table 11: Average LRU-normalized cost ratios on SPEC CPU2006 Benchmark using the FitF page predictor.

| Predictor | LRU | MARKER | PA. | PA.&M.$^D$ | PA.&M.$^R$ | GUARD&PA. |
|---|---|---|---|---|---|---|
| FitF page predictor | 1 | 0.970 | 1.006 | 0.851 | 0.947 | **0.816** |

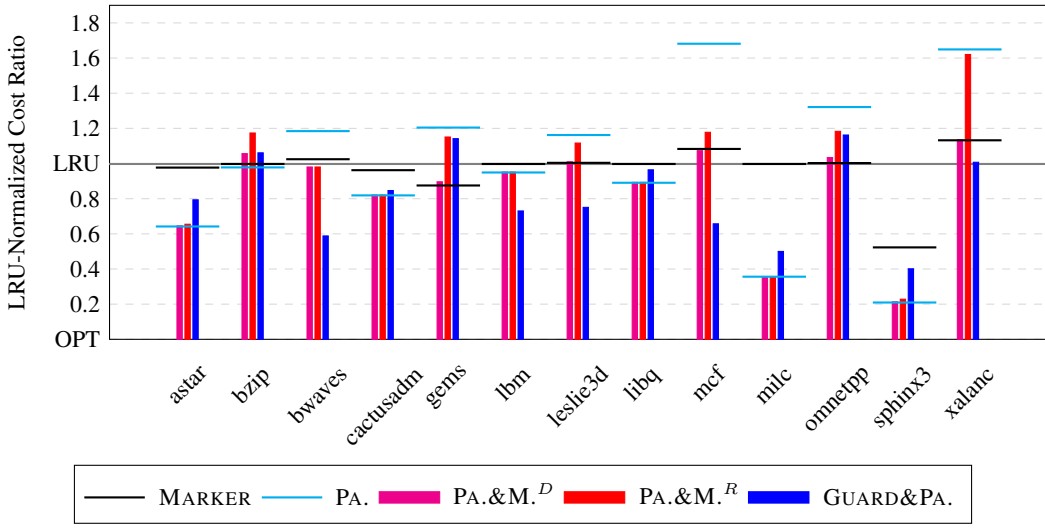

Figure 7: LRU-normalized cost ratios on SPEC CPU2006 Benchmark using the FitF page predictor.

# N    The EXGUARD Framework

## N.1    Motivation and Performance Overview

**Observation.** Essentially, the bounded robustness of GUARD&A stems from page protection, i.e., the guard mechanism. Its asymptotic logarithmic robustness depends on random evictions when necessary. However, overly aggressive random evictions may lead to a long eviction chain initiated by an eviction based on incorrect predictions, thereby degrading smoothness. Conversely, relying solely on predictions compromises robustness and may be impractical due to frequent predictor queries.

This motivates limiting the number of random evictions (Line 13 in Algorithm 10) on each eviction chain, as illustrated in Figure 8. This strikes a balance between smoothness and the number of predictions used without compromising the asymptotic robustness. Table 12 compares the applications of EXGUARD with those of GUARD.

Table 12: Robustified RB-following algorithms. Here, B.O. stands for BLINDORACLE. PA. denotes PARROT. Cons., #Pred. and Time. represent consistency, number of predictor calls, and time complexity, respectively. For EXGUARD, $d \in [1, H_k]$. For EXGUARD&B.O., $\lambda = h/e^h \leq 1/e$, where $h = H_k/(2d)$ and $h \in [1/2, H_k/2]$.

| Algorithm | Cons. | Robustness | Smoothness | #Pred. | Time. |
|---|---|---|---|---|---|
| GUARD&B.O. | 1 | $2H_{k-1} + 2$ | $\mathcal{O}(\log(\eta_t/\text{OPT}))$ | $\mathcal{O}(\text{OPT})$ | $\mathcal{O}(n \log k)$ |
| EXGUARD&B.O. | 1 | $2H_{k-1} + \mathcal{O}(d)$ | $\mathcal{O}(\min(\log(\frac{\eta_t}{\text{OPT}}), \lambda\sqrt{\frac{\eta_t}{\text{OPT}}}))$ | $\mathcal{O}(d \cdot \text{OPT})$ | $\mathcal{O}(n \log k)$ |
| GUARD&LRB | 1 | $2H_{k-1} + 2$ | $\mathcal{O}(H_k \cdot \eta_b/\text{OPT})$ | $\mathcal{O}(\text{OPT})$ | $\mathcal{O}(n)$ |
| EXGUARD&LRB | 1 | $2H_{k-1} + \mathcal{O}(d)$ | $\mathcal{O}(H_k/d \cdot \eta_b/\text{OPT})$ | $\mathcal{O}(d \cdot \text{OPT})$ | $\mathcal{O}(n)$ |
| GUARD&PARROT | 1 | $2H_{k-1} + 2$ | $\mathcal{O}(H_k \cdot \eta_f/\text{OPT})$ | $\mathcal{O}(\text{OPT})$ | $\mathcal{O}(n)$ |
| EXGUARD&PA. | 1 | $2H_{k-1} + \mathcal{O}(d)$ | $\mathcal{O}(H_k/d \cdot \eta_f/\text{OPT})$ | $\mathcal{O}(d \cdot \text{OPT})$ | $\mathcal{O}(n)$ |

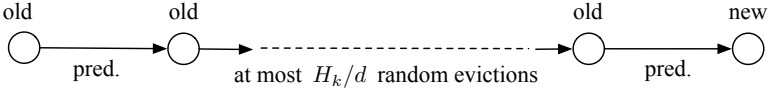

Figure 8: Random eviction budget along the eviction chain

**Algorithm 10** EXGUARD&A

1: $\mathcal{U} \leftarrow \emptyset$ (the set tracks *unrequested old pages* in the cache)
2: **for** $i = 1, ..., n$ **do**
3:     Receive a page request $r_i = (t_i, p_i)$ at time $t_i$
4:     **if** $p_i$ *is not in the cache* **then**
5:         **if** *the cache is full* **then**
6:             **if** $\mathcal{U}$ is empty **then**
7:                 *Unguard all cached pages*
8:                 $\mathcal{U} \leftarrow \{\text{all cached pages}\}$ *(a new phase begins)*
9:             **end if**
10:             $\mathcal{E} \leftarrow$ the corresponding eviction chain of $p_i$
11:             $B(\mathcal{E}) \leftarrow$ the random eviction budget of $\mathcal{E}$
12:             **if** $p_i$ *was evicted in the current phase* **then**
13:                 **if** $B(\mathcal{E}) \leq 0$ **then**
14:                     Call function Evict($\mathcal{U}$)
15:                     $B(\mathcal{E}) = H_k/d$
16:                 **else**
17:                     Evict a page $x$ from $\mathcal{U}$ uniformly at random
18:                     $B(\mathcal{E}) \leftarrow B(\mathcal{E}) - 1$
19:                 **end if**
20:                 *Guard* $p_i$
21:             **else**
22:                 $\mathcal{S} \leftarrow \{\text{all unguarded cached pages}\}$
23:                 Call function Evict($\mathcal{S}$)
24:                 $B(\mathcal{E}) = H_k/d$
25:             **end if**
26:             **if** the evicted page $x \in \mathcal{U}$ **then**
27:                 $\mathcal{U} \leftarrow \mathcal{U} \backslash \{x\}$
28:             **end if**
29:         **end if**
30:         Load $p_i$ into the cache
31:     **end if**
32:     **if** $p_i \in \mathcal{U}$ **then**
33:         $\mathcal{U} \leftarrow \mathcal{U} \backslash \{p_i\}$
34:     **end if**
35: **end for**

## N.2 Algorithm Descriptions

Algorithm 10 describes the EXGUARD framework. Algorithm 7, 8, and 9 can also be embedded into Algorithm 10 to generate the corresponding algorithms: EXGUARD&BLINDORACLE, EXGUARD&LRB, and EXGUARD&PARROT.

**Theorem N.1.** EXGUARD&A *is 1-consistent and* $2H_{k-1} + \mathcal{O}(d)$*-robust.*

*Proof.* We focus on the modification introduced by EXGUARD over GUARD.

Since EXGUARD&A retains the guard mechanism, it remains 1-consistency, as established by Proposition 4.1 and Corollary 3.7.

From the analysis in Appendix J, the expected length of an old-page eviction chain is at most $\mathcal{O}(H_k)$. EXGUARD introduces at most $\mathcal{O}(d)$ additional prediction-driven evictions along each chain, resulting in $(2H_{k-1} + \mathcal{O}(d))$-robustness. $\square$

## N.3 Predictor Usage Proof for EXGUARD

**Theorem N.2.** GUARD&A *and* EXGUARD&A *calls the predictor* $\mathcal{O}(\text{OPT})$ *and* $\mathcal{O}(d \cdot \text{OPT})$ *times, respectively, where* $A \in \{\text{BLINDORACLE}, \text{LRB}, \text{PARROT}\}$.

*Proof.* Recall that the predictor is called only when an eviction is required, consistent with common implementations in real systems. Let $A \in \{\text{BLINDORACLE}, \text{LRB}, \text{PARROT}\}$ in the following.

For GUARD&A, a prediction-driven eviction occurs only when serving a request for a new page. Thus, the number of predictor calls in phase $q$ is $n_q$. By Lemma 4.4 and 4.3, we bound the total number of predictor calls as:

$$\sum_{q \in \mathcal{Q}} n_q \leq \sum_{q \in \mathcal{Q}} 2c_q \leq 4\text{OPT} = \mathcal{O}(\text{OPT}). \tag{16}$$

EXGUARD&A introduces additional $o_q/(H_k/d)$ calls for the predictor. According to Theorem N.1, the expected total cost of EXGUARD&A satisfies $\sum_{q \in \mathcal{Q}} (n_q + o_q) \leq \text{OPT} \cdot \mathcal{O}(H_k)$. Therefore, the total number of predictor calls used by EXGUARD&A is bounded by:

$$\sum_{q \in \mathcal{Q}} (n_q + d \cdot o_q/H_k) \leq \mathcal{O}(d \cdot \text{OPT}). \tag{17}$$

(16) and (17) complete the proof. $\qquad\square$

## N.4 Smoothness Proof for EXGUARD&BLINDORACLE

**Theorem N.3.** EXGUARD&BLINDORACLE *is* $\mathcal{O}\big( \min(\log(\eta_t/\text{OPT}), h/e^h \cdot \sqrt{\eta_t/\text{OPT}})\big)$-*smooth, where* $h = H_k/(2d)$.

*Proof.* For an old-page eviction chain $\mathcal{E}_q$ in phase $q$ with $x$ prediction-driven evictions, the expected length of it is $|\mathcal{E}_q| \leq x \cdot H_k/d$. When there are $z$ unrequested old pages in the cache, the expected number of subsequent random evictions satisfies $L(z) \leq H_z \leq \ln(k) + 1$. (Refer to Appendix J for $L(z)$.) Thus, if a prediction-based eviction results in $y$ subsequent random evictions, it implies at least $\mathcal{O}(e^y)$ inversions in expectation, arising from all pairs between the prediction-driven evicted page and other unrequested old pages.

From the above, if $\mathcal{E}_q$ contributes $I(\mathcal{E}_q)$ inversions, then $I(\mathcal{E}) \geq \mathcal{O}(x^2 \cdot e^{H_k/d})$ since there are $x$ prediction-driven evictions. Therefore, $|\mathcal{E}_q| \leq x \cdot H_k/d \leq \mathcal{O}\left( \frac{H_k}{d} \cdot \sqrt{\frac{I(\mathcal{E}_q)}{e^{H_k/d}}} \right)$.

The total number of inversions satisfies $\text{inv}(\theta, \hat{\theta}) \geq \sum_{q \in \mathcal{Q}} \sum_{\mathcal{E}_q} I(\mathcal{E}_q)$. Let $N$ denote the total number of old-page eviction chains, where $\text{OPT} \leq N \leq 2\text{OPT}$, as shown in Equation (13) of Appendix K.2. We then bound the total cost of EXGUARD&BLINORACLE, denoted as $\mathbb{E}[\text{ExG\&B}]$, by

$$\mathbb{E}[\text{ExG\&B}] \leq \sum_{q \in \mathcal{Q}} c_q + \sum_{q \in \mathcal{Q}} \sum_{\mathcal{E}_q} \mathcal{O}\big(H_k/d \cdot \sqrt{I(\mathcal{E}_q)/e^{H_k/d}}\big) \quad \text{(by Lemma J.1)}$$

$$\leq \sum_{q \in \mathcal{Q}} c_q + N \cdot \mathcal{O}\Big(H_k/(de^{H_k/(2d)}) \cdot \sqrt{\frac{\text{inv}(\theta, \hat{\theta})}{N}}\Big) \quad \text{(by Jensen's inequality)}$$

$$= \mathcal{O}\Big(\text{OPT} \cdot h/e^h \cdot \sqrt{\frac{\eta_t}{\text{OPT}}}\Big), \quad \text{(by Lemma K.3)} \tag{18}$$

where $h = H_k/(2d)$. ExG.&B.O. inherits the $\mathcal{O}(\log(\eta_t/\text{OPT}))$ smoothness of GUARD&B.O., adding at most $\mathcal{O}(d)$ evictions per old-page chain, where $1 \leq d \leq H_k$. Combining the two bounds above completes the proof.

$\qquad\square$

## N.5 Smoothness Proof for EXGUARD&LRB

**Theorem N.4.** EXGUARD&LRB *is* $\mathcal{O}(H_k/d \cdot \eta_b/\text{OPT})$-*smooth.*

*Proof.* ExGuard&LRB limits the number of subsequent random evictions to $H_k/d$ after a prediction-based eviction. According to the smoothness analysis of Guard&LRB (Theorem K.6), and given a prediction error of $\eta_b$, the expected total cost of ExGuard&LRB is at most $\mathcal{O}(H_k/d \cdot \eta_b)$, thereby completing the proof. $\square$

## N.6 Smoothness Proof for ExGuard&Parrot

**Theorem N.5.** ExGuard&Parrot *is* $\mathcal{O}((H_k/d \cdot \eta_f/\text{OPT})$-*smooth.*

*Proof.* Similarly, ExGuard&Parrot limits the number of subsequent random evictions to $H_k/d$ after a prediction-based eviction. Based on the smoothness analysis of Guard&Parrot (Theorem K.7), we can bound the expected total cost of ExGuard&Parrot, denoted $\mathbb{E}[\text{ExG\&P}]$, as:

$$\mathbb{E}[\text{ExG\&P}] \leq \mathcal{O}(H_k/d \cdot \eta_f), \tag{19}$$

which completes the proof. $\square$

## N.7 Trade-offs Between Smoothness and Predictor Usage in ExGuard

Table 13: **Comparison of Smoothness and Predictor Usage**.

| NRT | Smoothness | #Pred. |
|---|---|---|
| P.M. | $\mathcal{O}(\sqrt{\varphi_t})$ | $\mathcal{O}(H_k \cdot \text{OPT})$ |
| LM. | $\mathcal{O}(\log(\varphi_t))$ | $\mathcal{O}(\text{OPT})$ |
| LNonM. | $\mathcal{O}(\log(k)/k \cdot \varphi_t)$ | *unbounded* |
| B.O. | $\mathcal{O}(1/k \cdot \varphi_t)$ | *unbounded* |
| B.O.&LRU | $\mathcal{O}(1/k \cdot \varphi_t)$ | $\mathcal{O}(k \cdot \text{OPT})$ |
| B.O.&M.$^D$ | $\mathcal{O}(1/k \cdot \varphi_t)$ | $\mathcal{O}(H_k \cdot \text{OPT})$ |
| B.O.&M.$^R$ | $\mathcal{O}(1/k \cdot \varphi_t)$ | $\mathcal{O}(H_k \cdot \text{OPT})$ |
| B.O.&EQ.$^R$ | $\mathcal{O}(1/k \cdot \varphi_t)$ | $\mathcal{O}(H_k \cdot \text{OPT})$ |
| G.&B.O. | $\mathcal{O}(\log(\varphi_t))$ | $\mathcal{O}(\text{OPT})$ |
| ExG.&B.O. | $\mathcal{O}(\min(\log(\varphi_t), \lambda\sqrt{\varphi_t}))$ | $\mathcal{O}(d \cdot \text{OPT})$ |

| Binary | Smoothness | #Pred. |
|---|---|---|
| M.P. | $\mathcal{O}(H_k \cdot \varphi_b)$ | $\mathcal{O}(H_k \cdot \text{OPT})$ |
| Mark0 | $\mathcal{O}(H_k \cdot \varphi_b)$ | *unbounded* |
| G.&LRB | $\mathcal{O}(H_k \cdot \varphi_b)$ | $\mathcal{O}(\text{OPT})$ |
| ExG.&LRB | $\mathcal{O}(H_k/d \cdot \varphi_b)$ | $\mathcal{O}(d \cdot \text{OPT})$ |

| Action | Smoothness | #Pred. |
|---|---|---|
| F&R ($f(i) = 2^i - 1$) | $\mathcal{O}(\log(\varphi_a))$ | $\mathcal{O}(\sqrt{k} \cdot \text{OPT})$ |
| F&R | $\mathcal{O}(f^{-1}(\varphi_a))$ | $\mathcal{O}(f(\log k) \cdot \text{OPT})$ |

| FitF | Smoothness | #Pred. |
|---|---|---|
| F&R (FitF) | $\mathcal{O}(\log(k)/b \cdot \varphi_f)$ | $\mathcal{O}(b \cdot \text{OPT})$ |
| G.&Pa. | $\mathcal{O}(H_k \cdot \varphi_t)$ | $\mathcal{O}(\text{OPT})$ |
| ExG.&Pa. | $\mathcal{O}(H_k/d \cdot \varphi_f)$ | $\mathcal{O}(d \cdot \text{OPT})$ |

**Notations in Table 13.** Abbreviations include BlindOracle (B.O.), PredictiveMarker (P.M.), LNonMarker (LNonM.), LMarker (LM.), BlindOracle&Marker (B.O.&M.), Equitable (EQ.), Mark&Predict (M.&P.), Parrot (Pa.), Guard (G.), and ExGuard (ExG.).

#Pred. represents the upper bound of the number of predictor calls. F&R's smoothness depends on $f(i)$, which limits predictor calls. Our experiments focus on the F&R variant with optimal smoothness ($f(i) = 2^i - 1$). For clarity, we denote $\eta_t/\text{OPT}$ as $\varphi_t$, $\eta_b/\text{OPT}$ as $\varphi_b$, $\eta_c/\text{OPT}$ as $\varphi_a$, and $\eta_f/\text{OPT}$ as $\varphi_f$, where $\eta_t, \eta_b, \eta_c, \eta_f$ represent different types of prediction errors.

For ExGuard, parameter $d \in [1, H_k]$. The smoothness parameter $\lambda$ of ExG.&B.O. satisfies $\lambda = h/e^h \leq 1/e$ (see Theorem N.3), where $h = H_k/(2d)$. For F&R (FitF), parameter $b \in \{1, ..., \log k\}$.

**Analysis of Predictor Usage.** B.O. and MARK0. suffer from unbounded robustness and invoke the predictor on each cache miss, resulting in unbounded predictor usage. The predictor usage of F&R and F&R (FITF) has been analyzed in the corresponding paper.

P.M. invokes the predictor $\mathcal{O}(H_k \cdot \text{OPT})$-times, as it evicts pages based on predictions whenever the corresponding eviction chain has length at most $H_k$. In contrast, LM. relies on predictions only once at the beginning of each eviction chain, resulting in $\mathcal{O}(\text{OPT})$ predictor queries. The predictor usage of LNONM. is unbounded, as it invokes the predictor once per eviction chain, and the number of eviction chains depends on the total prediction error.

M.&P. invokes the predictor on each cache miss, so its predictor usage bound is the product of $\mathcal{O}(\text{OPT})$ and its robustness. Similarly, the predictor usage of B.O.&A is bounded by the product of $\mathcal{O}(\text{OPT})$ competitive ratio of the classical algorithm A, where $A \in \{\text{LRU}, \text{MARKER}, \text{EQ.}\}$.

**Favorable Tradeoffs Achieved by EXGUARD.** For algorithms relying on NRT predictions, EXG.&B.O. achieves better tradeoffs than P.M. and LM. as their tradeoffs are theoretically directly comparable. To compare with other algorithms such as B.O.&LRU, we conduct experiments, and the results indicate that EXG.&B.O. achieves the best empirical tradeoff.

Moreover, both EXG.&B.O. and G.&B.O. exhibit good smoothness, even with $\mathcal{O}(\text{OPT})$ predictor usage. This enhances practicality, especially when predictor calls are costly.

Among algorithms that rely on binary predictions or the FitF predictor, both EXG.&LRB and EXG.&PA. achieve state-of-the-art tradeoffs.

## N.8 Experimental Results for EXGUARD

We empirically compare EXGUARD&B.O. with other algorithms, with $d$ set to $H_k$.

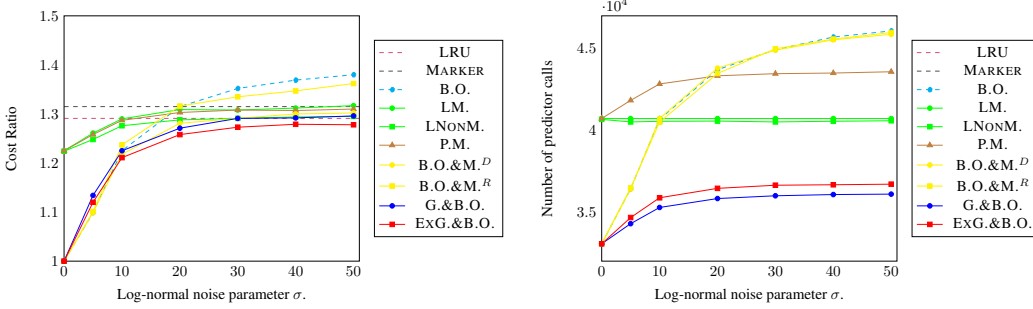

Figure 9: Performance with synthetic predictions of NRT on BrightKite.

Figure 10: Number of predictor calls with synthetic predictions of NRT on BrightKite.

Table 14: Average cost ratios ($\overline{\text{CR}}$), average LRU-normalized cost ratios ($\overline{\text{LCR}}$), and average number of predictor calls ($\overline{\text{#Pred.}}$) on SPEC CPU2006 Benchmark using PLECO predictor.

| Metrics | LRU | B.O. | P.M. | LM. | LNONM. | B.O.&M.$^D$ | G.&B.O. | ExG.&B.O. |
|---|---|---|---|---|---|---|---|---|
| $\overline{\text{CR}}$ | 1.478 | 1.404 | 1.335 | 1.335 | 1.346 | 1.294 | 1.226 | **1.225** |
| $\overline{\text{LCR}}$ | 1 | 1.049 | 0.946 | 0.942 | 0.951 | 0.766 | 0.627 | **0.623** |
| $\overline{\text{#Pred.}}$ | 0 | 79488 | 75870 | 72682 | 72767 | 79488 | **62099** | 62553 |

**Improved Performance.** Figure 9, 10 and Table 14 show that EXGUARD&B.O. outperforms GUARD&B.O. through increased predictor usage, empirically validating its improved smoothness. It is noteworthy that G.&B.O. and EXG.&B.O. depend less on the predictor than other algorithms utilizing next request time (NRT) predictions.

In the above, B.O.&M.$^R$ is excluded due to underperforming relative to B.O.&M.$^D$. F&R is excluded due to its reliance on action prediction. Although action predictions can be generated using (NRT), doing so requires NRT predictions for all requested pages. In contrast, other algorithms only predict the NRT of cached pages at eviction (predictions can be reused if not requested). Thus, their predictor call counts are not comparable.

# O   Other Caching Models

Beyond the classical caching model, where item sizes and cache-miss costs are uniform, researchers have also investigated more general variants such as weighted caching, in which different pages incur different loading costs, both without machine learning [47] and with learning-augmented approaches [48, 49]. However, research on learning-augmented algorithms for general caching settings, such as those with variable-sized items, remains limited and deserves further study.

In addition to accounting for cache-miss costs and item sizes, the conventional caching model inherently assumes that every requested item is stored upon access. This constraint is prevalent in classical systems. (1) In operating systems, virtual memory management depends on the guarantee that once a page is requested, it must be loaded into physical memory for execution to continue; otherwise, data errors, crashes, or inconsistent behavior may occur [50]. (2) In databases, this constraint ensures data consistency and transaction integrity in relational systems [51]. If a page is not retained in memory after being accessed until the transaction completes, it may result in partial updates or lost modifications, thereby violating the transaction's atomicity.

In contrast, the constraint can be relaxed in several modern settings. (1) In web browsers, caching is more flexible, as algorithms such as TinyLFU [52] explicitly allow the system to skip caching certain requested items. (2) In content delivery networks, caching decisions are governed by parameters like TTL, content popularity, and regional demand [53], meaning that not all requested content is cached immediately at the edge. In summary, whether this constraint can be relaxed depends on scenario-specific requirements. Future research could explore learning-augmented algorithms that operate without this constraint and investigate how cache admission choices impact robustness.

# P   Experimental Setup

Most of our experiments rely solely on a standard computer equipped with a CPU and RAM. Evaluating algorithms that use a neural network-based predictor, such as PARROT, requires a GPU.

# Q   Broader Impacts

This paper presents the design, analysis, and evaluation of learning-augmented algorithms for the caching (paging) problem. Our work has a potential positive societal impact by improving the efficiency and performance guarantees of caching systems.

