# OpenReview forum: "Robustifying Learning-Augmented Caching Efficiently without Compromising 1-Consistency"
_NeurIPS.cc/2025/Conference — NeurIPS 2025 poster_

### Official Review · Reviewer_Dovr · 2025-07-02

**Clarity:** 3
**Significance:** 2
**Originality:** 3
**Rating:** 4
**Confidence:** 1

**Summary:**

This paper addresses the online caching problem, focusing on minimizing cache misses under limited cache size. While existing learning-augmented caching algorithms achieve ideal 1-consistency, they often lack robustness guarantees. To tackle this, the authors propose GUARD, a lightweight robustification framework that significantly improves robustness to  2Hk+2 without sacrificing 1-consistency. GUARD maintains the original algorithm’s time complexity by adding only a constant per-request overhead. Extensive experiments on various real-world datasets and prediction models demonstrate the practical effectiveness of GUARD.

**Questions:**

see weakness

**Ethical Concerns:**

["NO or VERY MINOR ethics concerns only"]

**Quality:**

3

**Strengths And Weaknesses:**

Strengths:

1. The paper provides thorough and rigorous theoretical analysis
2. The writing is clear and well-structured.
3. This work is well-motivated.

Weaknesses:

1. The experiments could include more ablation studies to isolate the effects of different components of GUARD.
2. It is unclear how GUARD performs under extreme or adversarial conditions that may arise in some practical caching scenarios.
3. The paper could provide more detailed discussion on the potential limitations or failure cases of the method.

---

> ### Author Rebuttal · Authors · 2025-07-31
>
> Thank you for your comments and your support for our paper.
>
> **Weakness 1:**
>
> Guard has two main components at its core: protecting certain pages (Line 12 in Algorithm 1) and randomly evicting an unrequested old page (Line 11 in Algorithm 1) when a prediction error is detected (Line 10). We have conducted ablation experiments using variants of Guard that exclude random eviction or page protection. The table below extends the results of Table 7 in Appendix N.2, presenting average cost ratios across 13 SPEC CPU2006 benchmark datasets. The results indicate that both components are beneficial, with protection having a greater impact.
>
> | Predictor | LRU | Marker | B.O. | LM. | LNonM. | P.M. | B.O.\&M.^D | F\&R | Guard | Guard (without random eviction) | Guard (without protection) |
> |-----|-----|-----|-----|-----|-----|-----|-----|-----|-----|-----|-----|
> | PLECO  | 1.478  | 1.394  | 1.404 | 1.335  | 1.345  | 1.335 | 1.294 | 1.362 | 1.226  | 1.234  | 1.335 |
> | POPU   | 1.478  | 1.394  | 1.261 | 1.319  | 1.312 |  1.312 | 1.231 | 1.319 | 1.203  | 1.207 |  1.320 |
>
>
> **Weakness 2:**
>
> For learning-augmented algorithms, adversarial conditions include adversarial inputs to online problems (in our caching problem, the request sequence/trace) and adversarial predictions (extremely inaccurate predictions) in the learning-augmented setting.
>
> Theoretically, we have established the theoretical robustness of Guard, which directly corresponds to its worst-case performance bound under both adversarial traces and predictions. Its cost is at most $2H\_k + 2$ times that of the optimal.
>
> Empirically, it is challenging to reproduce both adversarial conditions for Guard or other existing learning-augmented algorithms, as real-world traces rarely exhibit truly adversarial behavior. As a result, previous work in this field, including the milestone work [1], often simulates only adversarial predictions in experiments. Similarly, in Section 5.2, we used synthetic predictors to evaluate the algorithms under varying levels of prediction error on a real-world dataset, BrightKite, and demonstrated that Guard performs robustly in these scenarios.
>
> [1] Thodoris Lykouris and Sergei Vassilvtiskii. Competitive caching with machine learned advice. Proceedings of the 35th International Conference on Machine Learning, 2018.
>
> **Weakness 3:**
>
> Thanks for pointing this out. Our framework is tailored to the standard paging problem, where all items have equal size and cost, and thus does not directly apply to more general settings like weighted or variable-sized caching. Like most learning-augmented caching algorithms, Guard also relies on high-performance predictors to be practical in low-latency systems, which makes reducing predictor usage important. As shown in the additional appendix (Appendix Q) provided in the supplementary material, Guard already reduces predictor usage compared to prior methods. Nonetheless, we believe that exploring the lower bound of predictor usage while maintaining strong performance would be an interesting direction for future work. We will add more discussion on limitations and future directions.

---

### Official Review · Reviewer_KGyM · 2025-07-03

**Clarity:** 3
**Significance:** 3
**Originality:** 3
**Rating:** 5
**Confidence:** 3

**Summary:**

The paper studies the online paging problem in the learning-augmented setting. It provides a framework that, given a consistent learning-augmented algorithm following a relaxed version of Belady's rule, maintains consistency while improving robustness to $2H_k+2$. That is, the final algorithm performs optimally for perfect predictions, and in the worst case, its cost is at most $2H_k+2$  times more than the optimal cost. The authors also perform extensive experiments to empirically compare the proposed algorithms with the baselines.

**Questions:**

N/A

**Ethical Concerns:**

["NO or VERY MINOR ethics concerns only"]

**Final Justification:**

The concerns raised by the reviewers appear to have been addressed by the authors. I will keep my original rating.

**Limitations:**

yes

**Quality:**

3

**Strengths And Weaknesses:**

1. Unlike the common approach in the framework of algorithms with predictions, which focuses on predicting the input, this work starts with a learning-augmented algorithm with specific properties and robustifies it without losing consistency. This is an interesting approach and may have applications in other problems studied in the learning-augmented setting.

2. They achieve a state-of-the-art consistency-robustness tradeoff.

3. The proposed robustification technique is simple, lightweight, and introduces minimal computational overhead, making it practical for real-world deployment.

4. The authors conduct comprehensive experiments that demonstrate performance improvements over existing baselines.

5. The paper is well-written and well-organized, making the technical content accessible and easy to follow. In particular, the detailed comparison with recent learning-augmented caching algorithms is helpful.

---

> ### Author Rebuttal · Authors · 2025-07-31
>
> Thank you very much for the review and for acknowledging our work.
>
> We are encouraged that our proposed robustification approach may be applicable to other problems, and that the comparison of existing learning-augmented caching algorithms is helpful. We hope this work can inspire and benefit other researchers in the community, help advance research at the intersection of machine learning, theoretical computer science, and computer systems, and support the application of machine learning in real-world systems efficiently and robustly. Any further suggestions are welcome.

---

### Official Review · Reviewer_tNoD · 2025-07-03

**Clarity:** 3
**Significance:** 2
**Originality:** 3
**Rating:** 5
**Confidence:** 4

**Summary:**

The paper investigates the caching problem, where the objective is to store $k$ pages to minimize cache misses---a scenario where a requested page is not present in the cache. The authors specifically examines learning-augmented caching policies, which leverage predictions to guide decisions about page retention and eviction. A key aspect of these policies is their performance under varying prediction accuracy: a consistent policy performs as well as the best caching algorithm when predictions are perfect, while a robust policy guarantees a bounded competitive ratio against the optimal solution even when predictions are flawed. The paper's primary contribution is the proposed algorithm GUARD, which serves as a wrapper to enhance the robustness of any learning-augmented algorithm without compromising its consistency.

**Questions:**

* Could you clarify the necessity of the system model's constraint that a specific cache must always store the requested file? What benefits or challenges might arise if this constraint were more flexible?

* How does the "guard" mechanism's complexity, which depends on the request sequence size (n), impact the practical scalability of your "lightweight" approach, especially with very large and frequent request sequences in real-world systems?

**Ethical Concerns:**

["NO or VERY MINOR ethics concerns only"]

**Final Justification:**

The authors satisfactorily addressed my concerns, leading me to raise my score from weak accept to accept.

**Limitations:**

yes

**Paper Formatting Concerns:**

The reviewer found no major formatting issues.

**Quality:**

3

**Strengths And Weaknesses:**

**Strengths.**

* The paper does a good job of situating its contributions within the relevant literature, and the research gap it seeks to fill is clearly articulated. The individual contributions are presented in a logical and easily understandable manner, and the work itself appears to be comprehensive and technically solid.

**Weaknesses.**

* It's unclear why the system model necessitates a specific cache always storing the requested file. While this aligns with the operation of many caching policies (e.g., LRU, LFU), its inclusion as a general system model constraint lacks clear justification. Allowing for greater flexibility in this regard could potentially lead to even stronger comparative optimums.

* The authors claim their approach is lightweight; however, the complexity of the "guard" mechanism appears to depend on the size of the request sequence (n). In real-world systems, where request sequences can be very large and frequent, it's not clear how this algorithm would remain practical.

---

> ### Author Rebuttal · Authors · 2025-07-31
>
> Thank you for your review and your valuable feedback. Since the weaknesses correspond to the questions, we respond to them together in detail.
>
> **Weakness and Question 1 (part 1):**
>
> Overall, we use the current system model for two main reasons as follows.
>
> First, this model captures a wide range of real-world systems, many of which explicitly require that the requested page or data be cached immediately after access as a hard constraint. For example, in operating system virtual memory management, the OS must load the requested page into memory and cache it in the page table or physical memory upon a page fault. This is essential for memory correctness and helps avoid costly disk I/O on repeated accesses. Similarly, in database buffer management for the most popular class of relational databases, a page is cached in the buffer pool and pinned as soon as it is read, such as during a transaction, because it must remain resident to ensure consistency and support rollback. A modern example can be found in GPU caching for recommendation models or large language models, where a vector such as an embedding or a KV cache vector must be cached in GPU memory after being fetched from host DRAM to support efficient reuse in subsequent forward passes.
>
> Second, since all existing learning-augmented caching algorithms, totaling more than 10 developed since 2018, use the current system model, we adopt the same model to ensure a fair performance comparison, as shown in Appendix B.
>
> **Weakness and Question 1 (part 2):**
>
> However, there are some scenarios where the requested item is typically cached after access, but this is not enforced as a hard constraint. Examples include applications such as web browsers, which cache page resources, and content delivery networks (CDNs), which store requested content at the edge. In these cases, there is an opportunity to relax this constraint, and we agree that doing so may improve caching performance.
>
> - **Benefits**: Without this constraint, the optimal algorithm can choose to skip storing the requested item if its next request time is later than that of all items currently in the cache. Essentially, this expands the eviction candidate set from $k$ to $k+1$, including the requested item as a candidate. This can reduce the optimal cost in certain cases, such as the following: suppose the cache initially contains $k$ items, $A_1$ to $A_k$. First, item $B_1$ is requested and will not be requested again. Next, items $A_1$ to $A_k$ are requested one by one. This two-step pattern is repeated for $n$ rounds, with $B_1, \ldots, B_n$ being requested in sequence. In this case, the optimal algorithm incurs a cost of $2n$ if it is required to store each requested item, but only $n$ if storing is optional. While the above case is extreme, it shows the potential benefits of relaxing the constraint. Similarly, learning-augmented algorithms may also benefit, particularly when the predictor is highly accurate.
>
> - **Challenges**: Incorporating this flexibility requires only minor modifications to existing learning-augmented algorithms, leaving their core structure unchanged. For example, in the case of Guard, we can forgo storing the requested item if its eviction priority (e.g., predicted next request time) is higher than that of all cached items. This modification can also be applied to other learning-augmented algorithms, making them easily adaptable to the new system model.
>
>     Therefore, adopting a more flexible system model may not pose significant challenges. The key challenge in this setting is ensuring that the algorithm avoids cases where frequently requested items are consistently not stored due to inaccurate predictions, leading to significant cost. We will outline the discussion of the new system model in the conclusion section.
>
> **Weakness 2 and Question 2:**
>
> Algorithm 1 presents the application of the *guard* mechanism to serve a sequence of $n$ requests. The mechanism introduces a total time complexity of $O(n)$ after serving $n$ requests, resulting in an amortized cost of $O(1)$ per request. This matches the overall overhead of classical caching algorithms such as LRU and LFU, which update the last access time or item frequency per request, also incurring a total cost of $O(n)$. In general, to robustify an online learning-augmented algorithm, the additional overhead mentioned above is inevitable. Existing learning-augmented mechanisms all incur at least additional $O(n)$ overhead over $n$ requests. For example, switching between algorithms in BlindOracle\&LRU incurs an $O(n)$ overhead, while the customized mechanism in F\&R incurs an $O(n^2)$ overhead after serving $n$ requests.
>
> Below, we list the time cost associated with each relevant line in Algorithm 1 when implemented in real-world systems.
>
> - (1) Upon receiving a request $r_i$ (Line 3), if the page is already in the cache, we set its *requested* tag of this page (Lines 23, 24), and decrease the number of unrequested old pages by 1, both in $O(1)$ time. If a cache miss occurs and the cache is full (Line 5), we check whether the number of unrequested old pages is zero (Line 6), also in $O(1)$ time. If so, a new phase begins: we remove the *guard* tag and *requested* tag of all cached pages (Line 7-8), which costs $O(k)$ in total, amortized to $O(1)$ per request.
>
> - (2) When a prediction error is detected (Line 10) via a hash table lookup, we add the *guard* tag to the requested page (Line 12), which costs $O(1)$. To perform random eviction from the set $U$ (Line 11), we use a temporary array of cache indices, i.e., $1, \ldots, k$, and randomly shuffle their order at the beginning of each new phase at a cost of $O(k)$, amortized to $O(1)$ per request. When a random eviction is needed, we scan the pages using shuffled indices and select the first one whose *requested* tag is unset. The next random eviction starts from the last page found and moves forward. This results in a worst-case cost of $O(k)$ per phase, amortized to $O(1)$ per request.
>
> - (3) In Line 14, pages with the
> *guard* tags are excluded from prediction. The complexity of the algorithm A's policy (Line 15) is independent of the *guard* mechanism.
>
> After being robustified by Guard, the new algorithms, including Guard\&B.O., Guard\&LRB, and Guard\&PA., achieve state-of-the-art asymptotic time complexity among learning augmented algorithms that use different types of predictions, as summarized in Table 4 of Appendix B.
>
> Overall, the *guard* mechanism maintains very low additional time complexity, asymptotically matching the time complexity of classical algorithms. In most scenarios, the overhead of classic caching algorithms like LRU is generally negligible compared to I/O or network latency, thereby ensuring practical scalability of Guard in real-world systems with large and frequent request streams.
>
> We hope this response addresses your concern and would be happy to engage in further discussion.

---

> > ### Comment · Reviewer_tNoD · 2025-08-08
> > **Responses to Points 1 and 2**
> >
> > The reviewer is now satisfied that Concern #2 has been adequately addressed, as it is now clearer that the amortized complexity is indeed O(1).
> >
> > Regarding Concern #1, it would be beneficial to include a more detailed explanation of the practical applications and scenarios in which the constraint in the system model is justified. Please also include relevant citations, and discuss under what circumstances this constraint can be relaxed. It would be valuable to add this discussion to the manuscript. Additionally, the phrase "may not pose significant challenges" is not sufficiently formal. If this aspect represents a limitation, the authors may consider stating it explicitly as such, or suggesting it as a direction for future work to encourage follow-up research based on this paper.

---

> > > ### Author Response · Authors · 2025-08-09
> > > **Thank you for the response (part 1)**
> > >
> > > We are glad to know our response has addressed Concern #2, and appreciate your additional suggestions related to Concern #1.
> > >
> > > **Additional Explanation:** We agree with you that it would be valuable and helpful to the reader to provide a more detailed explanation of scenarios where the constraint is justified or not, along with the reasons. The following references have been found to support this discussion, and more materials will be collected. An additional section will be added to the manuscript, organized to outline the scenarios. The initial draft is as follows.
> > >
> > > - Operating Systems: This constraint is required by the current system design of the modern OS. Particularly, virtual memory management relies on the assumption that once a page is requested, it must be loaded into physical memory for the process to continue [1, 2]. If the requested item were not loaded into the cache (e.g., DRAM), it could lead to data errors, crashes, or inconsistent behavior in the running application. Thus, this assumption necessitates the use of the current system model.
> > >
> > > - Databases: This constraint is required for data consistency and transaction integrity in relational databases [3, 4]. During a transaction, modifications to a page must remain persistent in the buffer pool until the transaction is complete to ensure that the database adheres to the ACID properties, particularly atomicity and consistency. If a page were evicted from memory before the transaction concludes, it could lead to partial updates, data corruption, or loss of modifications, violating the transaction's atomicity.
> > >
> > > - GPU Cache: This constraint is enforced in GPU caching because embedding vectors or KVCache fetch operations from DRAM are performed in batches. After the batch fetch, GPU streaming multiprocessors directly interact with GPU HBM, using the cached vectors for subsequent forward passes, such as in deep recommendation models and large language models. This requires the vectors to remain in GPU HBM. If evicted, the forward process cannot continue.
> > >
> > > - Web Browsers: This constraint can be relaxed in this scenario, since there are no fixed cache mode limitations. Specifically, browsers use HTTP headers like `Cache-Control` to manage caching behavior; for instance, setting `Cache-Control: no-store` instructs the browser not to cache the resource at all. TinyLFU [7] is an algorithm applicable to web browser caching that explicitly allows the system to choose not to cache a requested item.
> > >
> > > - Content Delivery Networks: This constraint can be relaxed in scenarios such as live media streaming and file delivery, where CDN caching decisions depend on factors such as TTL (time-to-live) settings, content popularity, and regional demand [8]. As a result, not all requested content is cached immediately at the edge.
> > >
> > > Overall, whether this constraint can be relaxed depends on scenario-specific requirements, including inherent system design, transaction integrity, and data processing flow.
> > >
> > > [1] Thomas Anderson and Mike Dahlin. Operating Systems: Principles and Practice, Volume 3: Memory Management.
> > >
> > > [2] Tanenbaum AS, Bos H. Modern operating systems. Pearson Education, Inc.; 2015.
> > >
> > > [3] Gray J, Reuter A. Transaction processing: concepts and techniques. Elsevier; 1992 Sep 30.
> > >
> > > [4] Silberschatz A, Korth HF, Sudarshan S. Database system concepts.
> > >
> > > [5] Yingcan Wei, Matthias Langer, Fan Yu, Minseok Lee, Jie Liu, Ji Shi, and Zehuan Wang. 2022. A GPU-specialized inference parameter server for
> > > large-scale deep recommendation models. In Proceedings of the 16th
> > > ACM Conference on Recommender Systems. 408–419.
> > >
> > > [6] Lianmin Zheng, Liangsheng Yin, Zhiqiang Xie, Chuyue Livia Sun,
> > > Jeff Huang, Cody Hao Yu, Shiyi Cao, Christos Kozyrakis, Ion Stoica,
> > > Joseph E Gonzalez, et al . 2024. Sglang: Efficient execution of structured language model programs. Advances in Neural Information Processing Systems 37 (2024), 62557–62583.
> > >
> > > [7] Einziger G, Friedman R, Manes B. Tinylfu: A highly efficient cache admission policy. ACM Transactions on Storage (ToS). 2017 Nov 17;13(4):1-31.
> > >
> > > [8] Fan Q, Li X, Li J, He Q, Wang K, Wen J. PA-cache: Evolving learning-based popularity-aware content caching in edge networks. IEEE Transactions on Network and Service Management. 2021 Jan 22;18(2):1746-57.

---

> > > > ### Author Response · Authors · 2025-08-09
> > > > **Thank you for the response (part 2)**
> > > >
> > > > **Future Work Directions:** Thanks for pointing out the problem with the phrase. We intended to mean that relaxing the constraint will present several challenges, but will not result in existing robustification methods (e.g., algorithm switching and Guard) being completely unavailable. However, we believe that these challenges, as listed below, are worth addressing. This could serve as a direction for future work, which we will describe in the conclusion section to stimulate follow-up research.
> > > >
> > > > - **Additional Worst Cases**: As mentioned in our last response, under relaxed constraint, we need to additionally ensure that the learning-augmented algorithm avoids cases where frequently requested items are consistently not stored due to inaccurate predictions, leading to significant cost and continuous cache misses.
> > > >
> > > >     Suppose the predictor is almost accurate, and there is only one item whose next request time is always inaccurately predicted as an extremely large value, and is requested constantly. In this case, if BlindOracle\&LRU is currently following the algorithm BlindOracle, which blindly follows the predictions, it will incur continuous cache misses due to not storing this item, as its predicted next request time is higher than others. Such continuous single-item cache misses may lead to potential system problems.
> > > >
> > > >     This case highlights the need for fine-grained detection of prediction errors by tracking algorithm behavior instead of simply comparing the past performance of algorithms and switching between algorithms. Based on Guard, other researchers could explore better methods that are also capable of detecting prediction errors in a fine-grained way and obtaining better robustness in the new system model.
> > > >
> > > > - **New Analysis Required**: Given a request sequence, the optimal cost may decrease with the relaxed constraint. The competitive ratios of classical algorithms, along with the consistency and robustness of existing robustification methods, need to be reanalyzed. Moreover, whether simply adopting existing methods to the new system model by treating the requested item as an additional eviction candidate will perform better than keeping the original methods in the new system model is worth analyzing.
> > > >
> > > >     Based on these analyses, it would be interesting to explore how strategies related to loading or not loading the requested item into the cache affect robustness. Since some applications and scenarios align with the relaxed constraint, and there may be more in the future, it is meaningful to compare the new performance bounds of different algorithms within the new system model.
> > > >
> > > > Thank you sincerely for the response and for providing constructive suggestions that help improve the manuscript.

---

### Official Review · Reviewer_exXN · 2025-07-05

**Clarity:** 4
**Significance:** 3
**Originality:** 3
**Rating:** 4
**Confidence:** 3

**Summary:**

The authors address the problem of enhancing the robustness of learning-augmented caching algorithms. They propose GUARD, a general framework designed to robustify caching strategies that adhere to a relaxed version of Belady’s rule. GUARD achieves a robustness guarantee of $​2H_k + 2$ while incurring minimal additional computational overhead. The authors conduct experiments on multiple datasets to verify the effectiveness of the GUARD framework.

**Questions:**

Please see weaknesses above.

**Ethical Concerns:**

["NO or VERY MINOR ethics concerns only"]

**Final Justification:**

I thank the authors for their response. My concerns have been adequately addressed. I will maintain my score.

**Limitations:**

yes

**Quality:**

3

**Strengths And Weaknesses:**

**Strengths.** The authors consider an important problem of robustifying online caching algorithms and develop a general framework that can be utilized to robustify learning-augmented caching algorithms that follow the relaxed Belady’s rule. The algorithm is intuitive, while the results presented are impactful. The paper is well written and easy to follow even for a non-expert. The presented experiments demonstrate the effectiveness of the proposed framework.

**Weaknesses.** The paper has only minor weaknesses; however, I provide a few concerns that may be worth addressing.

- There is an issue with the notation, in  Section 3, OPT is defined to be an algorithm. Then, in Section 4, the authors provide upper and lower bounds on OPT. Please correct.

- The discussion in observation 2 can be strengthened with some references or examples. Currently, the argument appears somewhat speculative and lacks rigor.

- The authors first present a weaker robustness result of $2 H_k + 6$ and provide its proof. Then, the authors present the improved result with a robustness guarantee of $2 H_k + 2$. What is the need to present two types of guarantees on robustness?

- It would be useful to provide some discussion on the proof of Theorem 4.6. Specifically, how the proof differs from Theorem 4.5.

- I believe the introduction could be strengthened by more clearly articulating the relevance and potential impact of the proposed algorithm and analysis for the machine learning community.

---

> ### Author Rebuttal · Authors · 2025-07-31
>
> Thank you for reviewing our paper. We appreciate your detailed evaluation of our work.
>
> **Weakness 1:**
>
> The cost of OPT on sequence $\sigma$, i.e., OPT($\sigma$), is abbreviated as OPT by omitting $\sigma$ for brevity, as mentioned in Section 2. However, based on your suggestion, we will restate this in Section 4 to make it clearer to the reader.
>
> **Weakness 2:**
>
> We will revise Observation 2 to: "Many switching-based methods, such as BlindOracle&LRU [7] and F&R [8], attempt to detect prediction errors during execution to decide when to switch between a prediction-based policy and a robust fallback strategy. A common method, as used by BlindOracle&LRU and BlindOracle&Marker [7], involves comparing the current total cost incurred by the prediction-based algorithm with that of the fallback algorithm. … Alternatively, some methods, such as F&R and F&R (FitF) [8], detect prediction errors by explicitly recomputing the optimal solution over the observed request sequence up to the point of a cache miss, and checking whether the optimal solution would also incur a miss. While this yields highly accurate error detection, its computational overhead is prohibitively high for real-time applications. The differences among prediction error detection strategies used in switching-based algorithms are discussed in detail in Appendix B (Design Principles)."
>
> This revision introduces additional examples of existing algorithms, explains their underlying mechanisms, and references the related discussion in Appendix B.
>
> **Weakness 3 and 4:**
>
> The proof of Theorem 4.5 ($2H_k + 6$) is first presented under three assumptions to show the $O(\log k)$-robustness, as it is concise, easy to understand, and largely follows the classical proof framework of the Mark algorithm. However, it is difficult to derive a tight robustness bound within this proof framework, as the required assumptions make the bound loose. Thus, we prove Theorem 4.6 ($2H_{k-1} + 2$) using a different technique (eviction graph) without relying on any assumptions, resulting in a tight bound. As suggested, we will include the above discussion to highlight the differences between the two proof methods and enhance clarity.
>
> **Weakness 5:**
>
> Thanks for this suggestion. While this paper focuses specifically on learning-augmented algorithms, which sit at the intersection of machine learning and theoretical computer science, the caching problem itself is widespread across computer systems. Our work highlights the potential and effectiveness of machine learning in enhancing the performance of traditional caching algorithms in modern systems. The field of machine learning for systems has grown rapidly in recent years, leading to many emerging research challenges. This trend was highlighted by Google’s keynote at ASPLOS 2025, a top conference in computer systems, where various research opportunities were discussed, including ML for compilers, memory management, cluster scheduling and resource allocation, network routing, and other system-level applications.
>
> However, the lack of robustness guarantees remains a key barrier to applying machine learning models in real-world, risk-sensitive systems such as cloud infrastructures, thereby limiting their potential benefits. Another real-world challenge lies in achieving robustness efficiently, which is also a key focus of our paper. We believe the insights from our lightweight robustification framework may help guide the safer and more reliable application of machine learning in caching and other decision-making scenarios.
>
> Beyond caching, there are many other scenarios where robust performance guarantees are highly desirable. These include reliable resource provisioning under uncertain demand patterns through load forecasting in power grid management, and the use of robust Markov Decision Processes (MDPs) or reinforcement learning to ensure safe operation in autonomous systems and industrial control. Similar needs also arise in cloud autoscaling and financial trading under volatile conditions.
>
> Overall, this line of work on learning-augmented algorithms, including ours, supports the deployment of machine learning models in real systems, further demonstrates the value of ML research, strengthens the impact of the ML community, and may lead to new research opportunities, such as developing system-oriented low-latency models and enabling the safer integration of ML into systems. Given the above, we will outline the key relevance and potential implications in the introduction.

---

> > ### Comment · Reviewer_exXN · 2025-08-07
> > **Thanks for the response**
> >
> > I thank the authors for their response. My concerns have been adequately addressed. I will maintain my score.

---

> > > ### Author Response · Authors · 2025-08-07
> > > **Thank you for reviewing our response**
> > >
> > > We are glad our response has addressed your concerns. Thank you again for your time and valuable suggestions.

---

### Decision · Program_Chairs · 2025-09-17

**Decision:**

Accept (poster)

**Comment:**

This work in the learning-augmented algorithms realm proposes a new method for improving the robustness of other learning augmented  caching algorithms by limiting the set of pages that can be evicted and sometimes overriding the algorithm's decisions. The reviewers agreed on the novelty, and appreciated the additional empirical experiments.